# RanBP2-dependent annulate lamellae drive nuclear pore assembly and nuclear expansion

Junyan Lin [1,2,3,4], Arantxa Agote-Aran[1,2,3,4,8], Yongrong Liao[1,2,3,4], Mehdi Cloarec[1,2,3,4], Leonid Andronov[1,2,3,4,8,9], Rafael L. Schoch [1,2,3,4,5], Paolo Ronchi [6], Victor Cochard[7], Rui Zhu[1,2,3,4,5], Erwan Grandgirard [1,2,3,4], Xiaotian Liu[1,2,3,4], Marianne Victoria Lemée[1,2,3,4,10], Charlotte Kleiss[1,2,3,4], Christelle Golzio[1,2,3,4], Marc Ruff [1,2,3,4,5], Guillaume Chevreux [7], Yannick Schwab [6], Bruno P. Klaholz [1,2,3,4,5] & Izabela Sumara [1,2,3,4] ✉

Nuclear pore complexes (NPCs) enable nucleocytoplasmic transport. While NPCs primarily localize to the nuclear envelope (NE), they also appear in cytoplasmic endoplasmic reticulum (ER) membranes called annulate lamellae (AL). Though discovered in the mid-20th century, AL's function and biogenesis remain unclear. Previously considered exclusive to embryonic and malignant cells, we find AL in somatic mammalian cells. Under normal conditions, AL store pre-assembled AL-NPCs that integrate into the NE, producing approximately one-third of newly formed nuclear pores and supporting nuclear expansion during G1. Upon pathological stimuli, AL transfer to the NE is impaired, leading to their cytoplasmic accumulation. RanBP2 (Nup358) is essential for AL biogenesis, with its phenylalanine-glycine repeats promoting AL-NPC scaffold oligomerization. ER-associated Climp63 (CKAP4) directs AL-NPCs to ER sheets and the NE. This AL-driven nuclear pore formation is complementary to the canonical routes, constituting a distinct NPC assembly pathway. Our work uncovers the biogenesis mechanism of AL and the nuclear function of this key cellular organelle.

Annulate lamellae (AL) are specialized subdomains within the endoplasmic reticulum (ER), characterized by a variable array of parallel-arranged layers. These assemblies feature a regular distribution of pore structures embedded in the membranes, which closely resemble nuclear pore complexes (AL-NPCs)[1,2]. AL were commonly observed in rapidly developing, differentiating, germ and malignant cells[3–11]. Despite their early discovery by McCullough in 1952[12] and designation by Swift in 1956[13], and subsequent reports describing AL in specific cell types[3,7,14–21], the universal and evolutionarily conserved function as well as direct biogenesis mechanisms of AL have remained enigmatic.

It could be hypothesized that AL might be utilized by rapidly growing cells to increase the pool of nuclear envelope (NE) NPCs (NE-NPCs) and sustain optimal levels of nucleocytoplasmic transport. This process is vital for normal cellular function, and abnormal expression

[1]Institute of Genetics and Molecular and Cellular Biology (IGBMC), Illkirch, France. [2]Centre National de la Recherche Scientifique (CNRS), UMR, Strasbourg, France. [3]Institut National de la Santé et de la Recherche Médicale (INSERM), Strasbourg, France. [4]Université de Strasbourg, Strasbourg, France. [5]Centre for Integrative Biology (CBI), Illkirch, France. [6]European Molecular Biology Laboratory, Electron Microscopy Core Facility, Heidelberg, Germany. [7]Université Paris Cité, CNRS, Institut Jacques Monod, Paris, France. [8]Present address: Fundamentals of Acoustic, Institute of Biochemistry, ETH Zürich, Zürich, Switzerland. [9]Present address: Department of Chemistry, Stanford University, Stanford, CA, USA. [10]Present address: Center for Molecular Biology of Heidelberg University (ZMBH), DKFZ-ZMBH Alliance, Heidelberg, Germany. ✉e-mail: sumara@igbmc.fr

or localization of nucleoporins (Nups), the building units of NPCs, have been observed in cancer and neurodegenerative disorders[22–25]. However, it is currently unknown if the Nups localization defects, often visualized by low-resolution microscopy techniques as foci or granules, are linked to the presence of AL and if AL regulate nuclear function under physiological conditions.

Here, we address this long-standing knowledge gap and uncover both the universal cellular function of AL and the direct mechanisms underlying their biogenesis. We demonstrate that AL are more abundant than previously recognized and are present in the cytoplasm of various somatic cells under normal physiological conditions. In normally proliferating cells, AL contain pre-assembled AL-NPCs that contribute to nuclear pore assembly and promote nuclear expansion during G1 phase, revealing a distinct NPC assembly pathway complementary to the two previously characterized routes[26]. Under pathological conditions or stress stimuli, this supply is disrupted, leading to the clustering and expansion of AL-NPCs in the cytoplasm. Importantly, the component of the NPC cytoplasmic filaments RanBP2 (also known as Nup358) is required for the formation of AL in normal cells and for their clustering under stress conditions. N-terminal unstructured phenylalanine-glycine (FG) repeat region of RanBP2 promotes the oligomerization of NPC scaffold components, leading to AL-NPC formation in the cytoplasm. In addition, we identify the ER-resident protein Climp63 (CKAP4) that turns out to ensure the proper localization of AL-NPCs to ER sheets and their integration into the NE. Disrupting these biogenesis mechanisms inhibits AL-NE merging, nuclear pore assembly and nuclear expansion during G1 phase, illustrating the essential role of AL in normal cycling cells.

## Results

### AL are present in diverse somatic cells

To determine whether cytoplasmic Nup foci observed in various stressed cells represent AL, we used cellular models that have linked microtubules (MTs), fragile X-related proteins (FXRPs; FXR1, FXR2, and FMRP), and ubiquitin-associated protein 2-like (UBAP2L) to aberrant Nup accumulation in the cytoplasm. Indeed, previous findings demonstrated an important role of dynein-mediated MT-based transport as well as chaperoning proteins FXRPs and UBAP2L in the localized assembly of Nups at the NE during interphase[25,27,28]. Nup foci were also seen in the context of a human disease such as fragile-X syndrome (FXS)[25], which is characterized by the absence of FMRP and in several other human disorders[14]. Although FXRPs-UBAP2L models do not capture all perturbations that generate cytoplasmic Nup foci, they provide a well-established framework for assessing whether these structures correspond to AL.

As expected, acute (90 min) MT depolymerization or deletion of UBAP2L in mCherry-Nup133 knock-in (KI) HeLa cells (Supplementary Fig. 1a–d) led to the formation of cytoplasmic mCherry-Nup133 foci that also contained Nup62 (Supplementary Fig. 1e). Correlative light and electron microscopy (CLEM) analysis showed these foci to be localized to parallel-aligned ER membrane structures containing stacked nuclear pore-like complexes, suggesting that they represent AL (Fig. 1a).

To overcome existing limitations in the microscopy analysis of AL, we utilized single molecule localization microscopy[29,30] (SMLM) to generate super-resolution images of the AL foci. In particular, we applied a multi-color variant based on a dichroic image splitter and spectral demixing (splitSMLM)[31]. Via the simultaneous imaging of multiple species of fluorophores, the method enables high-precision observations of the structural organization of NPCs and other multi-component complexes, reaching a resolution of 20 nm[31]. This approach is easily applicable under various experimental conditions and can provide greater details on the distribution and composition of AL. We confirmed the presence of highly organized cytosolic assemblies containing tightly packed NPC structures, likely corresponding to

AL, upon disruption of the FXRPs-UBAP2L pathway (Fig. 1b, Supplementary Fig. 1f–h).

Surprisingly, small AL containing fewer NPCs (on average 1.65) could be observed in the cytoplasm of wild-type (WT) HeLa cells (Fig. 1b, c, Supplementary Fig. 1i), whereas in cells treated with the MT depolymerizing agent nocodazole AL were larger (average of 13.8 NPCs) (Fig. 1c). Accordingly, when imaged with conventional (diffraction-limited) fluorescence microscopy, AL foci also appeared smaller for WT HeLa interphase cells (average size 0.090 μm²) compared to cells with depolymerized MTs (average size 0.213 μm², Supplementary Fig. 2a-b). Similar small cytoplasmic Nup foci were previously postulated to represent AL in normal interphasic cells[32]. We observed a relatively broad distribution range of AL foci sizes (Supplementary Fig. 2a, b) in both WT and nocodazole-treated cells, confirming the reported diversity of AL morphologies and abundance[33]. Hereafter, AL foci considered "large" have a size bigger than 0.3 μm².

Importantly, small AL could be detected not only in a variety of common cancer cell lines including U2OS, DLD-1, HCT116, but also in the non-transformed cell lines such as RPE-1, FB789 and MRC5 (Supplementary Figs. 3a, 4a). Likewise, small AL structures were observed in normally proliferating human fibroblasts, while large AL were found in FXS patient-derived fibroblasts (Fig. 1d). Finally, AL structures were identified in human induced pluripotent stem cells (hiPSCs)-derived neurons (Fig. 1e, Supplementary Figs. 3a, 4a), corroborating our findings on the widespread existence of AL in somatic cells. Although AL abundance varied across different cell types (Fig. 1f), AL size consistently increased following nocodazole treatment (Supplementary Figs. 3b, 4b). These results illustrate the strength of splitSMLM as a powerful method to study AL and reveal the widespread existence of small AL in somatic, normally proliferating mammalian cells.

### AL cluster in the cytoplasm upon pathological stimuli

Small and large AL-NPCs were composed of most of the tested Nups that form different subcomplexes, with the exception of the nuclear basket nucleoporin Nup153 and ELYS, under various analyzed conditions (Fig. 1g, Supplementary Fig. 2c–f). Interestingly, RanBP2 was localized symmetrically on both sides of AL-NPCs (Supplementary Fig. 1f), consistent with previous reports[28,34]. In Drosophila embryos, AL-NPCs contain ELYS, but not Nup153 and Nup62[7] and in Xenopus egg extracts, AL-NPCs did not contain ELYS, while Nup153 and Nup62 were identified[35,36]. These results suggest that AL consist of pre-assembled NPCs, displaying compositional variations across different species. Nucleocytoplasmic transport factors Exportin-1, Importin β, and Ras-related nuclear protein (Ran) likewise localized to AL foci (Supplementary Fig. 5a), as previously reported[16], suggesting that they may be involved in AL-mediated NPC assembly, analogous to the established roles of these factors in NPC assembly pathways[10,36]. NE-resident proteins SUN1 and SUN2, but not Lamin A, Lamin B1, Nesprin, Emerin, and Lap2b, co-localized with AL foci (Supplementary Fig. 5b).

Since AL exist in various sizes and acute MT disruption leads to large AL, we studied their possible dynamic nature. Spinning disk confocal live video microscopy of WT mEGFP-Nup107 HeLa cells, synchronized in interphase, revealed a progressive increase in size and cytoplasmic clustering of AL foci over time following MT depolymerization (Supplementary Fig. 5c and Supplementary Video 1). Similar clustering events were observed upon depletion of FXR1 or in UBAP2L knockout (KO) cells (Supplementary Fig. 5d, e). These results demonstrate that small AL present in normal cells form large AL and accumulate in the cytoplasm when exposed to specific stimuli or in pathological states such as FXS (Fig. 1d).

### AL are dynamic and fuse with the NE during G1

The existence of small AL in normal cells prompted us to investigate their physiological role in normally proliferating cells and to address why pathological stimuli, such as MT disruption, lead to the

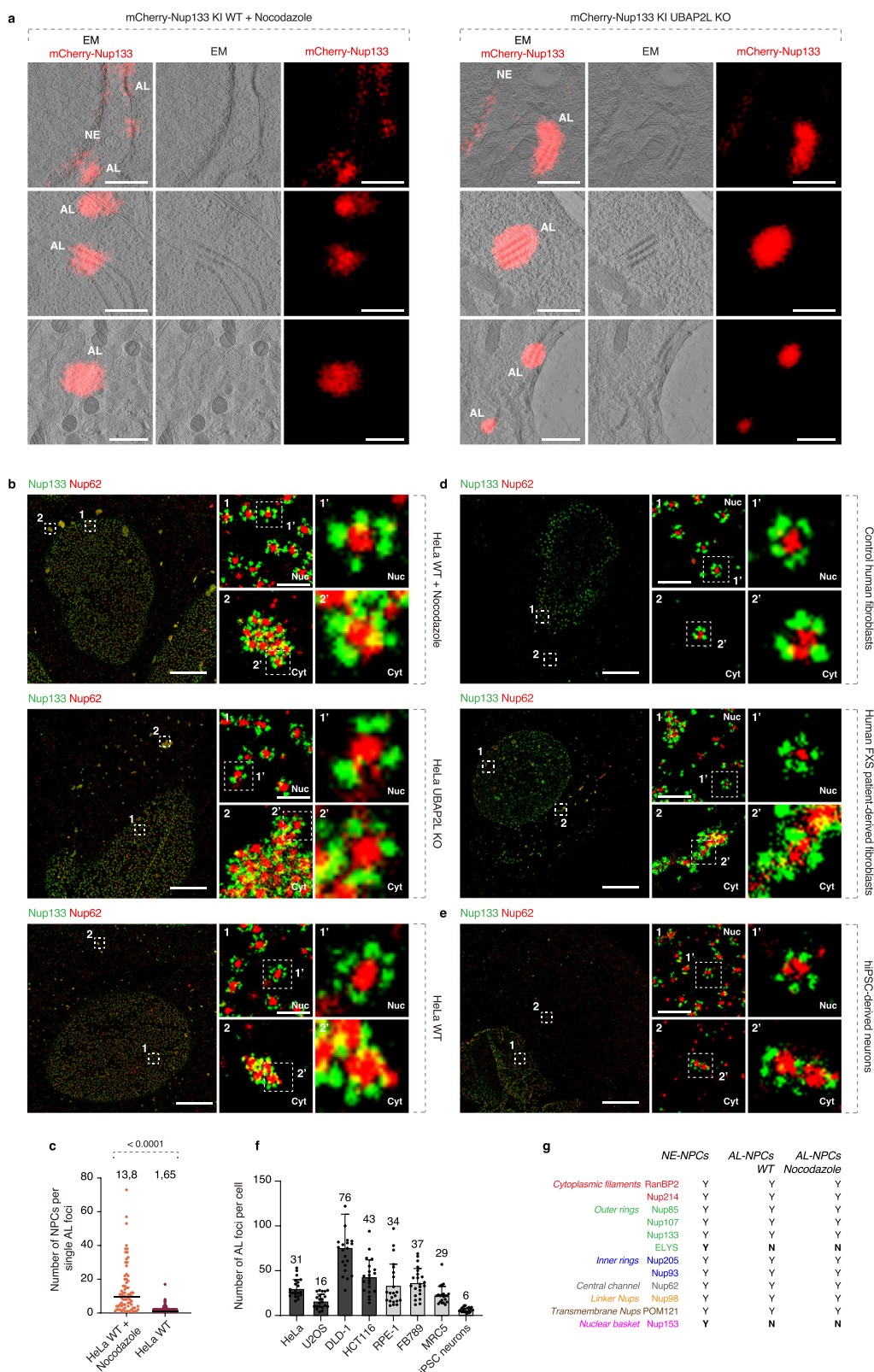

cytoplasmic accumulation of AL. While the conserved role of AL remains unclear, early studies in *Drosophila* embryos demonstrated that AL integrate into the NE to supply additional membranes and NPCs to support rapid nuclear growth, representing the first indication of the functional role of AL in the context of development[7]. (Fig. 2a, b, Supplementary Fig. 6a–d). Large AL foci induced by acute MT disruption followed the same cell cycle distribution pattern

(Supplementary Fig. 6e–j). It is therefore possible that nuclear growth during the G1 phase of somatic cells is supported by AL, which primarily exist during G1 (Fig. 2a, b, Supplementary Fig. 6a–d) and which could contribute to the incorporation of new NPCs into the NE.

A quantitative analysis of fixed cells estimated that AL-NPCs within the cytoplasmic AL account for approximately 3% of total nuclear NPCs, increasing to 10% following microtubule depolymerization (Fig. 2c),

**Fig. 1 | AL are present in diverse somatic cells. a** Representative correlative light and electron microscopy (CLEM) images of HeLa cells ($N = 3$). Left and right panels show nocodazole-treated cells (10 μM, 90 min) and UBAP2L knockout (KO) cells, respectively. mCherry fluorescence accumulates at densely packed nuclear pore complexes (NPCs) on stacked ER sheets in the cytoplasm corresponding to annulate lamellae (AL), and is also observed at the NE. Scale bars, 1 μm. **b** Representative splitSMLM images of NPCs at the nuclear (Nuc) surface and AL-NPCs in the cytoplasm (Cyt) of HeLa cells treated with nocodazole (10 μM, 90 min), UBAP2L KO HeLa cells, and WT HeLa cells ($N = 3$). Nup133 labels the cytoplasmic and nuclear rings, and Nup62 marks the central channel. AL-NPC clusters in WT cells are smaller than in other conditions. Boxed regions are shown at higher magnification in numbered panels. Scale bars, 3 μm (whole nuclei) and 0.3 μm (zoom).**c** Quantification of single AL-NPC complexes per AL foci/cluster. A total of 320 foci were analyzed (two-tailed unpaired t-test; mean ± SD; $N = 3$). Source data are provided as a Source Data file. **d** Representative splitSMLM images of NPCs at the NE and AL-NPCs in the cytoplasm of normal human fibroblasts and FXS patient-derived fibroblasts ($N = 3$). Boxed regions are shown at higher magnification in numbered panels. Scale bars, 3 μm (whole nuclei) and 0.3 μm (zoom). **e** Representative splitSMLM images of NPCs at the NE and AL-NPCs in the cytoplasm of hiPSC-derived neurons ($N = 3$). Boxed regions are shown at higher magnification in numbered panels. Scale bars, 3 μm (whole nuclei) and 0.3 μm (zoom). **f** Quantification of AL foci per cell in cancer (dark gray) and non-transformed (light gray) cell lines. At least 25 cells per line were analyzed (mean ± SD, $N = 3$). Source data are provided as a Source Data file. **g** Summary of the NE-NPC and AL-NPC composition. Colors indicate NPC subcomplexes; Y, present; N, absent.

potentially challenging this hypothesis. However, a simple static comparison of the NPC numbers in two compartments cannot accurately determine the contribution of AL-NPC to the NE. A more time-resolved analysis of the putative dynamic nature of AL structures is required.

For this reason, we used spinning disk live-cell imaging with short time intervals between the acquisition points. We found that AL foci are highly dynamic under physiological conditions, undergoing frequent fusion and fission events (Fig. 2e, Supplementary Video 2, 3), unlike the continuous fusion observed upon microtubule depolymerization (Supplementary Fig. 5c and Supplementary Video 1). Notably, small AL foci fused more rapidly with the NE than large AL (33 and 192 s, respectively), and a subset of small AL originated through the fragmentation of large AL and subsequently fused with the NE (81 s; Fig. 2d, e, and Supplementary Video 4). After contacting the NE, AL occasionally undergo multiple cycles of detachment and reattachment before ultimately fusing completely with the NE (Fig. 2d, e, and Supplementary Video 5).

In unstressed cells, small AL structures contain an average of 1.65 AL-NPCs (Fig. 1c). Given the observed net AL-NPC fusion frequency with the NE (14 events per 15 min; Fig. 2d), we estimate that approximately 924 AL-NPCs are incorporated into the NE during a single G1 phase of the cell cycle (10 h). Considering that the total number of nuclear pores in HeLa cells ranges from 2000 to 4000[37], the contribution of AL may in fact be substantial, accounting for an estimated 22–45% of total nuclear NPCs.

To provide additional evidence for the AL-dependent nuclear delivery of NPCs, we generated photoconvertible mEos2-Nup133 KI HeLa cells (Supplementary Fig. 6k–m). The mEos2 is a green-to-red photoconvertible fluorescent protein; although its green fluorescent state lacks sufficient photostability for long-term imaging[38]. Conversion of mEos2-Nup133 at the NE from green to red fluorescence at a defined interphase time point revealed no detectable diffusion of red signal from the NE into the cytoplasm (Supplementary Fig. 6n, and Supplementary Video 6). Interestingly, photoconversion of cytoplasmic mEos2-Nup133-positive AL foci resulted in the progressive accumulation of a weak but detectable red signal at the NE (Fig. 2f, g, and Supplementary Video 7), indicating that AL-NPCs can be incorporated into the nucleus. The weak nuclear signal is likely due to progressive photobleaching of the fluorescence signal in the course of the experiment and dynamic remodeling of AL foci, which flattened and dispersed as a rim around the nuclei after migration towards the NE (Fig. 2f). Future development of photoconvertible tags with improved long-term stability and brightness will be required to investigate the dynamics of AL incorporation into the NE in live cells in greater detail. Nevertheless, taken together our data show that AL structures observed in normally proliferating cells can attach to the NE during G1.

## AL-NPCs promote nuclear pore formation and nuclear expansion

To provide morphological evidence for AL incorporation into the NE, we next used splitSMLM to examine the structural organization and composition of AL-NPCs. This analysis visualized symmetric AL-NPCs in the proximity of or directly attached to the NE containing asymmetric NE-NPCs in WT interphase cells (Fig. 3a panels 1–2). AL-NPC fusion events into the NE were observed where symmetric AL-NPCs were positioned laterally onto the NE (Fig. 3a panels 2–4). Some fusion events were accompanied by the formation of lateral NE openings containing both AL-NPCs and NE-NPCs next to each other (Fig. 3a panels 5–6).

Analysis of Lamin A and B1 by splitSMLM revealed that despite the presence of these NE incisions, the underlying nuclear lamina remained intact (Fig. 3b). Importantly, the analysis of the ER marker PDI confirmed that the sites of AL-NPC insertion within these NE openings colocalized with the ER and ER-NE contact sites (Fig. 3c). Thus, the splitSMLM analysis confirms the observed AL structures in normal cells as specialized sub-compartments of the ER, which is known to be continuous with the outer NE and contribute to its formation[39]. These observations are also in agreement with a previously proposed model suggesting "en bloc" insertion of AL into the NE and remodeling of AL-NPCs by GTPase Ran to become asymmetric functional NPC structures in fly embryos[7,10,34], demonstrating evolutionary conservation of this process. Although 20 nm resolution images from splitSMLM analysis strongly suggest insertion of intact AL into NE, at this point we cannot formally exclude the possibility that AL-NPCs undergo local disassembly or that AL act solely as donors of NPC subcomplexes prior to their NE insertion.

Importantly, spinning disk confocal live video microscopy of the ER marker mScarlet-ER confirmed that AL foci move along the ER to merge with the NE in WT interphase cells (Fig. 3d, and Supplementary Video 8). MT depolymerization is predicted to modulate ER dynamics[40], damaging reticulated ER tubules and leading to vesiculation and thickening of cisternal sheets (Supplementary Fig. 7a), which could explain the inhibition of AL foci transfer to the NE and their clustering (Fig. 3d, Supplementary Fig. 5c and Supplementary Video 9). Interestingly, AL-NPCs were also often localized near microtubules (Supplementary Fig. 7b). Live-cell imaging further revealed a consistent spatial association between AL and the microtubule network (Supplementary Video 10), suggesting that they may utilize MT-ER interactions to merge with the NE and to increase the pool of NE-NPCs. Our results indicate that MT depolymerization in somatic cells disrupts ER dynamics, thereby impairing the transfer of AL-NPCs to the NE, resulting in their cytoplasmic accumulation. However, we cannot exclude the possibility that the transport of soluble Nups along microtubules also contributes to AL-NPC assembly. In support of this, studies in *Drosophila* oocytes have shown that MT mediate the delivery of distinct Nup condensates to AL-NPC assembly sites a process essential for efficient AL-NPC formation[10].

Since acute MT depolymerization inhibited AL-NPC transfer to the NE, we set out to analyze the physiological consequences of this inhibition. The density of NE-NPCs (Fig. 3e, f) and the level of Nups at NE (Supplementary Fig. 7c, d) decreased upon acute nocodazole treatment. The short treatment excluded any possible indirect effects

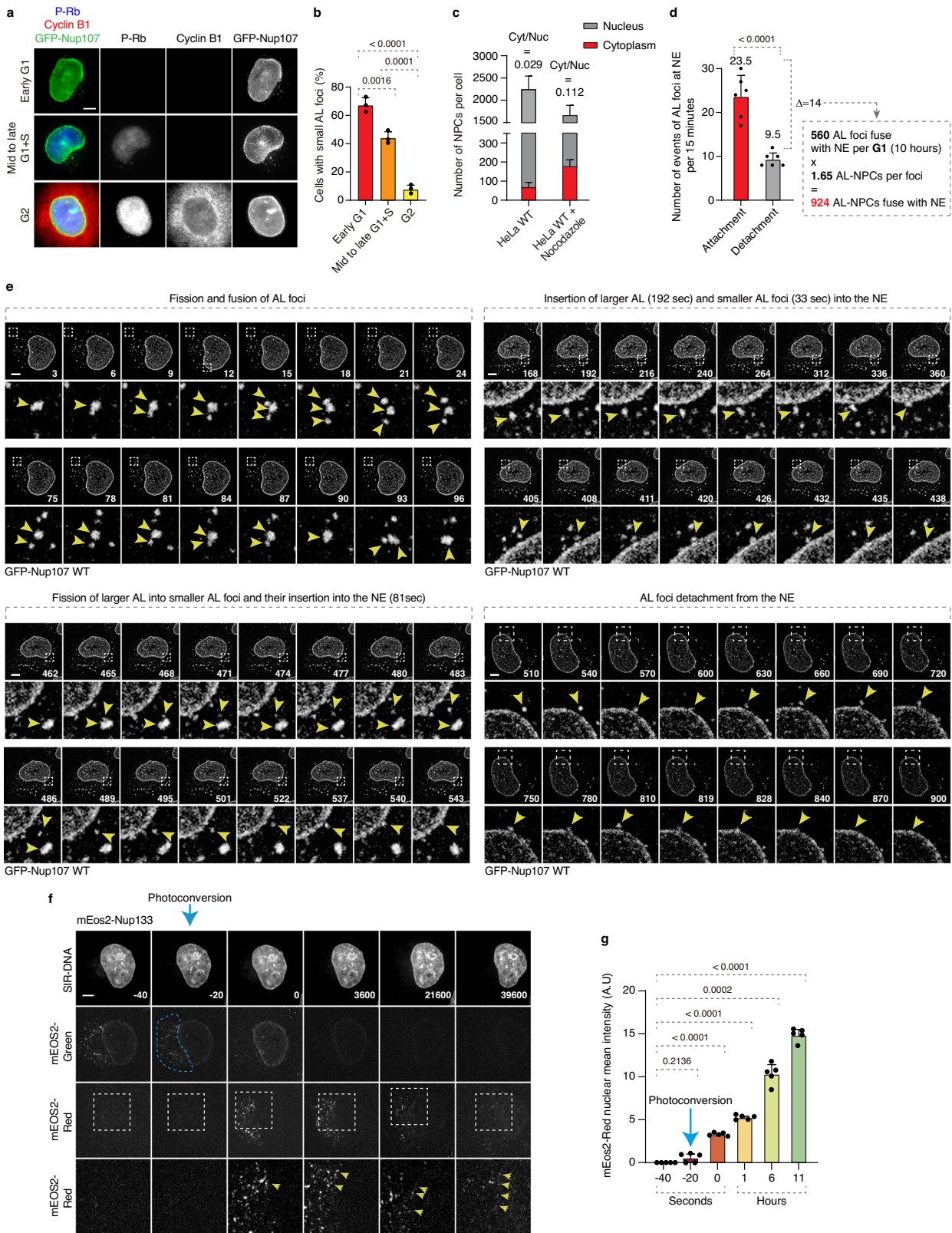

of nocodazole on mitotic progression. Strikingly, when cells are arrested in the G1/S phase by sustained addition of thymidine, the nucleus continues to grow beyond its normal size, and MT depolymerization blocks this nuclear growth in G1/S-arrested WT cells (Fig. 3g–j). Nuclear expansion is sustained by a constant supply of many proteins and lipids from the ER to the NE[41], a process which could also be disrupted by MT depolymerization. The insertion of AL-NPCs

may support nuclear expansion by potentiating nucleocytoplasmic transport rates, or the supply of lipids and proteins required for NE expansion may occur simultaneously with AL insertion into the NE. Taken together, AL are more widespread than previously anticipated and exist in WT somatic cells. AL act to supply a relatively large pool of AL-NPCs (estimated 924) to the NE, thereby promoting the nuclear expansion specifically during G1 phase of the cell cycle. AL are inserted

**Fig. 2 | AL are dynamic and fuse with the NE during G1. a**, **b** Representative images of asynchronously proliferating 2xZFN-mEGFP-Nup107 HeLa cells co-labeled with anti-p-Rb (blue) and anti-cyclin B1 (red) antibodies (**a**). The percentage of cells with Nup foci in p-Rb- and cyclin B1-negative cells (early G1), p-Rb-positive and cyclin B1-negative cells (mid- to late G1 and S phase), and p-Rb- and cyclin B1-positive cells (G2) was quantified (**b**). At least 200 cells per condition were analyzed (one-way ANOVA; mean ± SD; $N = 3$). Scale bars, 5 μm. Source data are provided as a Source Data file. **c** Quantification of NPC numbers in the cytoplasm and nucleus of individual cells. 6 cells per condition were analyzed (mean ± SD, $N = 3$). Source data are provided as a Source Data file. **d**, **e** Dynamics of AL foci in 2xZFN-mEGFP-Nup107 HeLa cells in G1 phase were analyzed by live spinning disk confocal microscopy. Selected representative frames are shown, with time indicated in seconds. Magnified boxed regions are shown in the lower panels. Yellow

arrowheads indicate the movement of cytosolic AL-NPCs. Scale bars, 5 μm. Attachment and detachment events of AL foci at the nuclear envelope (NE) were quantified (**d**). 6 cells were analyzed (two-tailed unpaired t-test; mean ± SD; $N = 3$). Source data are provided as a Source Data file. **f**, **g** Photoconversion of Nup foci in mEos2-Nup133 HeLa cells was analyzed by live spinning-disk confocal microscopy. Selected representative frames are shown, with time indicated in seconds. Magnified boxed regions of the mEos2-Red signal are shown in the lower panels. The blue dashed box indicates the photoconversion region, where mEOS2-Nups convert from green to red fluorescence. Yellow arrowheads indicate accumulation of AL-Nup133 at the NE. Scale bars, 5 μm. The average nuclear mEOS2-Red intensity over time was quantified (**g**). Five cells per condition were analyzed (one-way ANOVA; mean ± SD; $N = 3$). Source data are provided as a Source Data file.

into the NE at the ER-NE junctions through ER dynamics, and this process requires an intact microtubule network. These findings not only establish AL as functional determinants of nuclear architecture during normal mammalian cell proliferation but also indicate existence of an additional specialized pathway for NPC assembly during normal cell cycle progression.

## RanBP2 is required for clustering of AL under stress

Having demonstrated a cellular function of AL in normally growing cells, we next aimed to uncover the mechanisms driving AL biogenesis. The component of NPC cytoplasmic filaments RanBP2 (also known as Nup358) has been shown to localize to AL structures and it has been proposed to regulate a Nup condensate fusion mechanism supporting AL biogenesis during *Drosophila* oogenesis[10]. However, direct evidence for RanBP2's role in AL formation is missing. It is also unknown if the role of RanBP2 is conserved in somatic cells and in other species, and what the molecular basis for possible RanBP2 function on AL is. RanBP2 was required for the formation of large AL foci induced by MT depolymerization (Fig. 4a–c, Supplementary Fig. 7e–l) and by inactivation of FXRPs-UBAP2L-dynein pathway in several human cell lines tested (Supplementary Fig. 7m–r). Another component of NPC cytoplasmic filaments, Nup214, was not required for the formation of large AL foci (Supplementary Fig. 8a–c). Higher eukaryotes utilize two NE-NPC assembly pathways, each associated with different cell cycle stages: the postmitotic and the interphase pathway[26]. We found that downregulation of ELYS (required for postmitotic NE-NPC assembly), or Nup153 or POM121 (driving interphase NE-NPC assembly) did not inhibit AL foci formation when induced by acute nocodazole treatment Supplementary Fig. 8a–f. In contrast, defects in NE-NPC biogenesis were reported to induce AL[24] and as expected, downregulation of ELYS or Nup153 led to the formation of large AL foci in untreated cells (Supplementary Fig. 8a, d, g). Downregulation of RanBP2 strongly inhibited the formation of large AL foci induced by Nup153 or ELYS depletion (Fig. 4d–i), suggesting its universal and widespread role in AL clustering.

Is RanBP2 also required for the maintenance of the AL structures? Downregulation (Supplementary Fig. 9a–f) or auxin-inducible depletion (Supplementary Fig. 9g–k) of RanBP2[42] after nocodazole treatment severely decreased the number of large AL foci and their clustering observed upon MT depolymerization. Thus, RanBP2 is required for the formation and the maintenance of large AL. This process is likely to be independent of reported NPC degradation mechanisms involving autophagy[43,44], since RanBP2 downregulation inhibited large AL foci formation also in the presence of lysosomal inhibitors (Supplementary Fig. 10a–h).

Previous studies have shown that RanBP2 knockdown reduces translation of reporter constructs[45], whereas we show that deletion or downregulation of RanBP2 did not alter the protein levels of multiple Nups across various cell lines (Supplementary Fig. 10i–k). Notably, the use of auxin-inducible RanBP2 depletion minimizes the confounding effects of chronic translational defects (Supplementary Fig. 10k).

Therefore, the observed reduction in AL is unlikely to result from global translational regulation of Nups by RanBP2 under these conditions. The RanBP2 function on AL could also not be explained by possible effects on levels of FXR1 and UBAP2L, the factors driving spatial assembly of Nups during early interphase[28], as they were not affected by auxin-inducible deletion of RanBP2 in DLD-1 (Supplementary Fig. 10l) and in HCT116 cells (Supplementary Fig. 10m). Thus, RanBP2 is required for the clustering of small AL into large AL and for AL maintenance but does not regulate protein levels of Nups or Nup-interacting factors.

## RanBP2 supports small AL biogenesis and nuclear function in G1

Is RanBP2 also required for the formation and function of small AL that we identified under normal growing conditions? The splitSMLM analysis revealed a relatively broad distribution of the number of NPCs in a single AL-NPC structure (Fig. 4j–l) and as expected, acute MT depolymerization increased both the number of individual AL-NPCs in single AL foci and the total number of AL-NPCs per cell relative to WT cells (Fig. 4j–l). Importantly, RanBP2 was required for the formation of individual AL-NPCs, as well as their clustering both in WT and in nocodazole-treated cells (Fig. 4j–l). AL-NPCs were reported to lack nuclear basket components[7,35,36], whereas RanBP2 appears to be localized symmetrically on both sides of AL-NPCs[28] (Fig. 3a, b, Supplementary Fig. 1f) and RanBP2 can oligomerize to form multimers[45], providing a possible explanation for its ability to cluster individual AL-NPCs. In accordance with its role in AL-NPC biogenesis, RanBP2 was required for the formation of small AL foci in WT cells from G1 to S phase but not in the G2 stage (Fig. 4m, Supplementary Fig. 11a) and RanBP2 downregulation by siRNA decreased levels of Nups at the NE (Supplementary Fig. 11b, c). Small AL foci and the levels of Nups at the NE during interphase also decreased following auxin-inducible deletion of RanBP2 (Supplementary Fig. 11d–h), suggesting that RanBP2-mediated formation of small AL is a prerequisite for their merging with the NE and increasing the pool of NE-NPCs. Short-term depletion of RanBP2 reduced levels of several Nups in isolated nuclear fractions and increased their abundance in the cytoplasm in DLD-1 (Fig. 4n) and HCT116 cells (Fig. 4o), consistent with the idea that RanBP2 does not regulate the total protein levels of Nups (Supplementary Fig. 10i–k) but their localization to the NE. Indeed, RanBP2 deletion resulted in reduced NE-NPC density (Fig. 4p, r).

Can RanBP2 also contribute to the nuclear pore function? We analyzed the nucleocytoplasmic distribution of endogenous Ran, as previously reported[28,46], focusing specifically on Ran protein localization rather than the nucleotide-dependent Ran-GTP gradient. Ran is actively imported into the nucleus by transport factors and exported following cargo release, allowing continuous cycling between nuclear and cytoplasmic compartments. Downregulation of RanBP2 (Supplementary Fig. 12a, b) increased the nuclear-to-cytoplasmic (N/C) ratio of Ran, consistent with impaired nuclear export of Ran protein. Acute depletion of RanBP2 similarly increased the N/C ratio of Ran, with

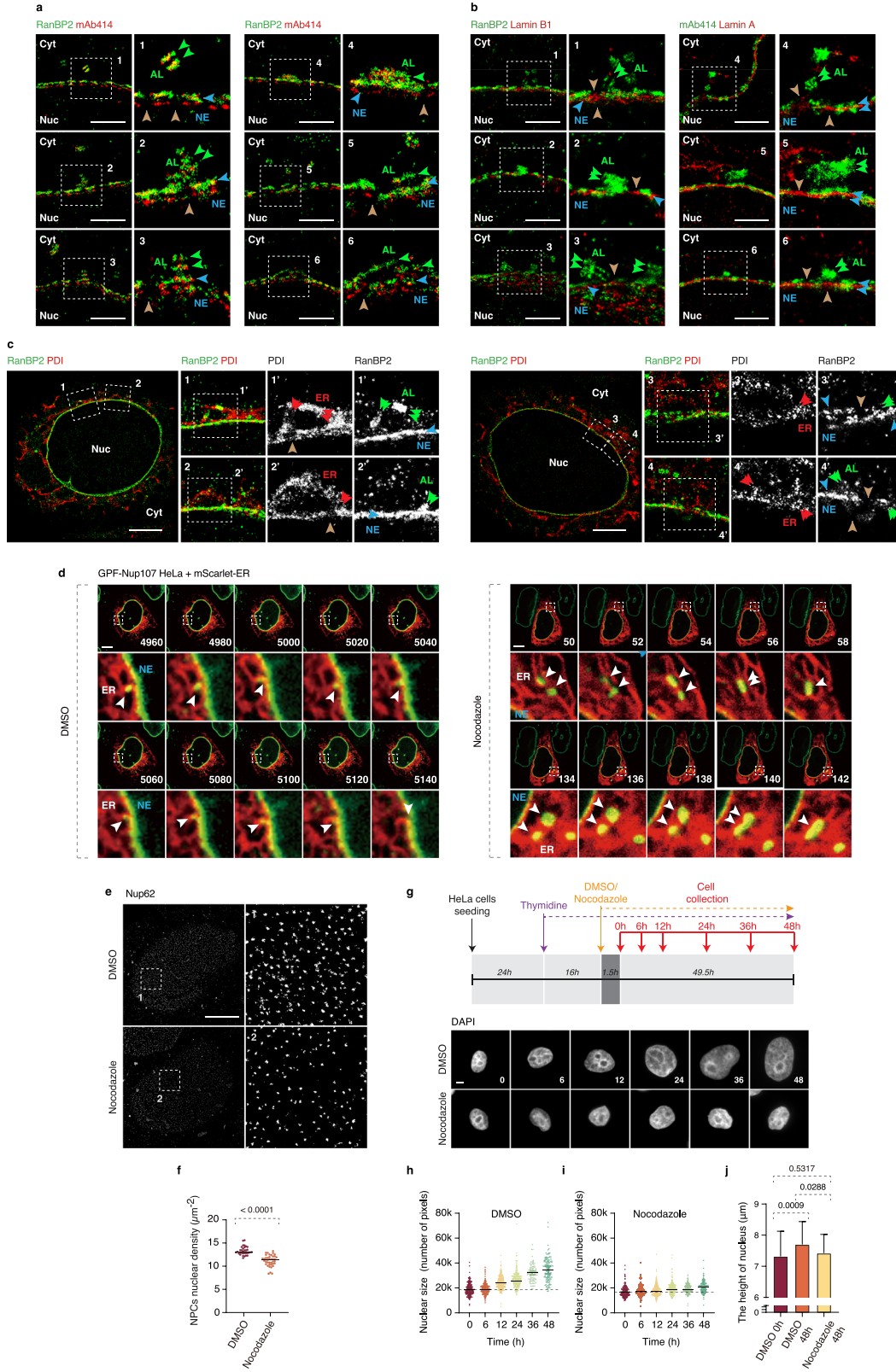

stronger effects observed in G1 compared to G2 cells (Supplementary Fig. 12c, e), suggesting that AL- and RanBP2-dependent NPC assembly contributes to efficient Ran export during early interphase. In contrast, prolonged RanBP2 depletion led to more pronounced defects in G2 cells (Supplementary Fig. 12d, e), likely reflecting cumulative perturbations in nucleocytoplasmic transport. Importantly, we note that the canonical Ran-GTP gradient is determined not solely by Ran

localization but by the coordinated activities of RCC1, RanGAP, and the nucleotide state of Ran. Moreover, the Ran-specific transport factor NTF2, which has been implicated in nuclear size scaling[47,48], was not directly examined here. Whether cell cycle-dependent regulation of these components contributes to the observed changes in Ran distribution remains an open question. Consistent with impaired Ran export, the use of a light-inducible nuclear export system (LEXY)[49]

**Fig. 3 | AL-NPCs promote nuclear pore formation and nuclear expansion.**
**a** Representative splitSMLM images showing the nuclear (Nuc) surface and cyto-
plasm (Cyt) from a side view of WT HeLa cells ($N = 3$). The central channel, cyto-
plasmic filaments, and nuclear basket of nuclear pore complexes (NPCs) are labeled
with mAb414 antibodies, and NPC cytoplasmic filaments are labeled with a RanBP2
antibody. Magnified boxed regions are shown in the corresponding numbered
panels. Green arrowheads indicate AL-NPCs within annulate lamellae (AL), where
NPC cytoplasmic filaments are symmetrically distributed, and blue arrowheads
indicate asymmetric NE-NPCs at the nuclear envelope (NE) (panel 1). Brown arrows
indicate NE openings. Lateral fusion events of AL-NPCs with NE-NPCs from the
cytoplasmic side are shown in panels 2–4, and NE openings containing both AL-
NPCs and NE-NPCs are shown in panels 5–6. Scale bars, 1 µm. **b** Representative
splitSMLM images showing the nuclear (Nuc) surface and cytoplasm (Cyt) from a
side view of WT HeLa cells ($N = 3$). NPC cytoplasmic filaments are labeled with a
RanBP2 antibody, and NE-associated factors are labeled with Lamin B1 and Lamin A
antibodies. The central channel, cytoplasmic filaments, and nuclear basket of NPCs
are labeled with mAb414 antibodies. Magnified boxed regions are shown in the
corresponding numbered panels. Green arrowheads indicate symmetric AL-NPCs
within AL, and blue arrowheads indicate asymmetric NE-NPCs at the NE. Brown
arrows indicate NE openings where NE-associated factors remain intact. Scale bars,
1 µm. **c** Representative splitSMLM images showing the nuclear (Nuc) surface and

cytoplasm (Cyt) from a side view of WT HeLa cells ($N = 3$). NPC cytoplasmic fila-
ments are labeled with a RanBP2 antibody, and the endoplasmic reticulum (ER) is
labeled with PDI. Magnified boxed regions are shown in the corresponding num-
bered panels. Red arrows indicate ER, green arrows indicate AL-NPCs, and brown
arrows indicate NE openings. Scale bars, 3 µm. **d** 2xZFN-mEGFP-Nup107 HeLa cells
expressing the ER marker mScarlet-ER were analyzed by live spinning-disk confocal
microscopy. Selected representative frames are shown, with time indicated in
seconds. Magnified boxed regions are shown in the lower panels. White arrowheads
indicate movement of cytosolic AL-NPC foci along the ER and their incorporation
into the NE at ER-NE junctions. Scale bars, 5 µm. **e**, **f** Representative SMLM immu-
nofluorescence images of Nup62 at the nuclear surface in DMSO- or nocodazole-
treated (10 µM, 90 min) HeLa cells (**e**). Magnified boxed regions are shown in the
right panels. Nuclear NE-NPC density (Nup62) was quantified (**f**) (two-tailed
unpaired t-test; mean ± SD; 40 cells per condition; $N = 3$). Scale bars, 5 µm. Source
data are provided as a Source Data file. **g**–**j** Schematic of the experimental setup (**g**).
HeLa cells were arrested in S phase by sustained addition of thymidine, treated with
nocodazole (10 µM), and fixed at the indicated time points. Representative images
of nuclei are shown, and nuclear area (**h**, **i**) and nuclear height (**j**) were quantified.
For nuclear area, 200 cells per condition were analyzed, and for nuclear height, 80
cells per condition were analyzed (one-way ANOVA; mean ± SD; $N = 3$). Scale bars,
5 µm. Source data are provided as a Source Data file.

confirmed export defects upon acute RanBP2 depletion (Supplemen-
tary Fig. 12f, g).

Our data in normally proliferating cells demonstrated a role for AL
in supplying the NE-NPC pool, thereby supporting the nuclear expan-
sion during early interphase. Since RanBP2 was required for AL-NPC
formation in WT cells, we next analyzed the role of RanBP2 in nuclear
function. Importantly, short-term depletion of RanBP2 was sufficient
to reduce nuclear size (Fig. 4q, s), expanding our observations in
nocodazole-treated cells (Fig. 3g–i). We conclude that RanBP2-
mediated AL-NPC assembly can preserve nuclear pore density as well
as nuclear growth during interphase, confirming the critical role of
cytosolic NPCs in the nuclear function. Collectively, RanBP2 plays
direct and widespread roles in AL, being essential for the biogenesis of
single AL-NPCs, their clustering and AL maintenance in multiple
stressed and normally proliferating somatic cells.

## RanBP2 FG repeats mediate AL assembly and nuclear function
Next, we set out to study the precise molecular mechanisms of
RanBP2-mediated AL biogenesis. First, to understand if the function of
RanBP2 on AL is specific and which functional domains of this large
protein (Fig. 5a) are required for AL formation, we performed rescue
experiments using RanBP2 fragments. The full-length (FL; aa 1-3224)
and the N-terminal (NT) fragment of RanBP2 (aa 1-1171) efficiently
reversed the inhibition of nocodazole-induced large AL foci observed
upon downregulation of RanBP2, and the FG repeat region (832-1171)
was essential for the RanBP2 AL clustering function (Fig. 5a–e). The
effects of auxin-inducible deletion of RanBP2 on AL formation could
also be efficiently rescued by the NT fragment of RanBP2 (aa 1-1171) but
not by the version lacking the FG repeat region (1-832; Supplementary
Fig. 13a–e).

Importantly, full-length RanBP2 with a deletion of the FG region,
which is not expected to affect other cellular functions of RanBP2,
failed to restore large AL foci in RanBP2-depleted cells by either auxin-
inducible degron (Fig. 5f–i) or RanBP2-specific siRNAs (Supplementary
Fig. 13f–i), confirming an essential role of the RanBP2 FG repeats in AL
accumulation. In contrast, mutation of the LIQIML motif, which is
important for RanBP2 targeting to the NE and its function in NE-NPCs
assembly[45], inhibited efficient NE localization of full-length RanBP2
leading to its nuclear and cytoplasmic localization, as expected, but
successfully rescued AL defects observed under both RanBP2 deletion
strategies (Fig. 5f–I, Supplementary Fig. 13f–i). These results further
support the conclusion that the FG region of RanBP2 regulates AL

formation independently of other known cellular functions of this
protein and identify a mutant form of RanBP2 which can separate its
functions at the NE and on AL.

These findings enabled us to further assess the role of AL in
nuclear expansion in untreated WT cells. The full-length WT and the
RanBP2 LIQIML mutant, but not the full-length RanBP2 lacking the FG
region, successfully rescued the nuclear expansion defect caused by
RanBP2 depletion by auxin-inducible degron or siRNA (Supplementary
Fig. 13j–l, Fig. 5j, k). Taken together, these results demonstrate the
important and specific role of the FG region of RanBP2 in AL formation
and AL-driven nuclear function.

## RanBP2 N-terminus drives AL-NPC oligomerization and assembly
To corroborate the role of the NT region of RanBP2, we set out to study
its molecular interactions relevant for AL biology. Ectopic expression
of NT RanBP2 (aa 1-1171) (Fig. 6a, b, Supplementary Fig. 14a) but not of
FL Nup85 or FL Nup133 (Supplementary Fig. 14b, c), induced AL foci
where multiple Nups, with the exception of ELYS and Nup153 (Fig. 6c),
could co-localize, relative to controls. CLEM analysis (Fig. 6d) and
splitSMLM (Fig. 6e) suggested that these foci represent AL. Ectopic
expression of NT RanBP2 increased the number of individual AL-NPCs
per cell and the size of AL (Fig. 6e–g). We conclude that elevated levels
of the NT RanBP2 fragment can induce AL under normal growth
conditions.

Since RanBP2 was required for the assembly of small AL foci and
individual AL-NPCs in the cytoplasm (Fig. 4j–l), we next asked how
RanBP2 NT regulates this crucial function at the molecular level.
Interestingly, structural analysis of the NPC demonstrated that the
N-termini of five RanBP2 molecules and one Nup93 molecule bind to
the overlapping region of two Y complexes (also named Nup107-
Nup160 complexes) in the scaffold's outer ring[45], and deletion of
RanBP2 resulted in an outer ring that lacked Y complexes on the
cytoplasmic side[50]. To test if RanBP2 can promote the oligomerization
state of scaffold components of AL-NPCs in a similar fashion, we ana-
lyzed separated cytoplasmic and nuclear cellular fractions by gel-
filtration chromatography (Supplementary Fig. 14d). Deletion of
RanBP2 inhibited the formation of oligomers of Y complexes and
Nup96-labeled scaffold in the cytoplasm but not in the
nucleus (Fig. 6h).

Intrinsically disordered FG repeats have been reported to stabilize
subcomplexes within NE-NPCs[51], therefore we examined if similar

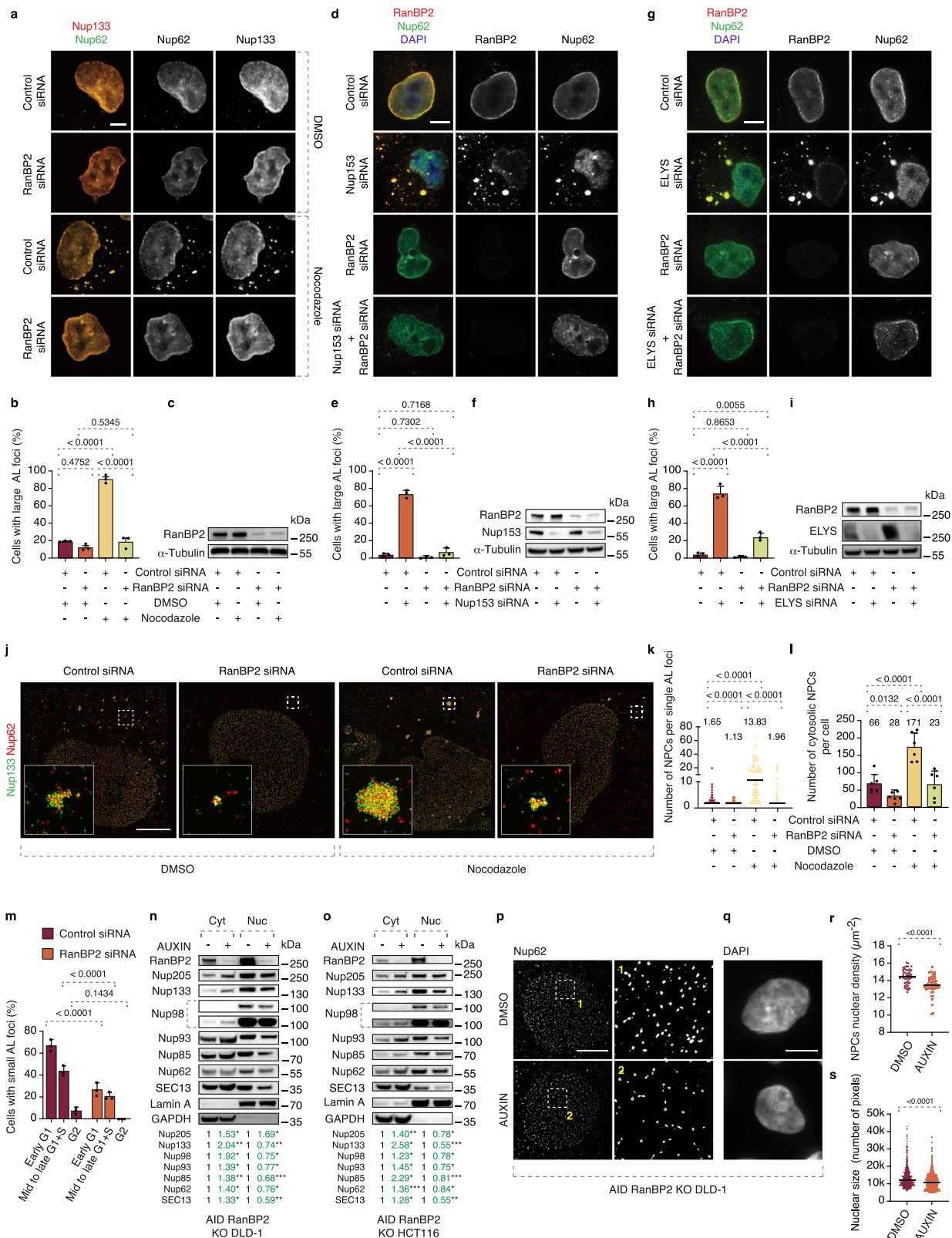

interactions between the RanBP2 NT fragment (aa 1-1171) and other Nups could play a role in AL-NPC assembly. The outer ring of the NPC is mainly composed of the evolutionarily conserved Y complex. We found that the N-terminal fragment of RanBP2 could bind the Y complex, as well as Nup205 and Nup93 (Fig. 6i), which belong to the inner ring and have recently been reported to also reside on the cytoplasmic face of NPCs[45]. The interaction of these Nups with NT RanBP2 was

dependent on the FG repeat domain (Fig. 6i), suggesting that it facilitates the assembly of single AL-NPCs. Deletion of RanBP2 did not affect the formation of the Y complex (Supplementary Fig. 15a) both in the cytoplasm and in the nucleus (Supplementary Fig. 15b), but it restricted the binding of the Y complex to Nup93 (Fig. 6j, Supplementary Fig. 15a) specifically in the cytoplasm (Supplementary Fig. 15c). This suggests that RanBP2 may stabilize interactions between

**Fig. 4 | RanBP2 is required for AL biogenesis, clustering and nuclear pore formation. a–c** Representative images of HeLa cells treated with the indicated siRNAs for 48 h and subsequently with nocodazole (10 µM) or DMSO for 90 min. Cells were co-labeled with anti-Nup62 (green) and anti-Nup133 (red) antibodies to detect AL-NPC foci (**a**). The percentage of cells containing large AL foci was quantified (**b**); at least 1000 cells per condition were analyzed (one-way ANOVA; mean ± SD; $N = 3$). Scale bars, 5 µm. Western blot analysis is shown in (**c**). Source data are provided as a Source Data file. **d–f** Representative images of HeLa cells treated with the indicated siRNAs for 48 h. Cells were co-labeled with anti-Nup62 (green) and anti-RanBP2 (red) antibodies to detect AL-NPC foci (**d**). The percentage of cells containing large AL foci was quantified (**e**); at least 300 cells per condition were analyzed (one-way ANOVA; mean ± SD; $N = 3$). Scale bars, 5 µm. Western blot analysis is shown in (**f**). Source data are provided as a Source Data file. **g–i** Representative images of HeLa cells treated with the indicated siRNAs for 48 h. Cells were co-labeled with anti-Nup62 (green) and anti-RanBP2 (red) antibodies to detect AL-NPC foci (**g**). The percentage of cells containing large AL foci was quantified (**h**); at least 300 cells per condition were analyzed (one-way ANOVA; mean ± SD; $N = 3$). Scale bars, 5 µm. Western blot analysis is shown in (**i**). Source data are provided as a Source Data file. **j–l** Representative splitSMLM images of HeLa cells treated with the indicated siRNAs for 48 h and subsequently with nocodazole (10 µM) or DMSO for 90 min, showing AL-NPCs in the cytoplasm. Cells were co-labeled with anti-Nup62 (green) and anti-Nup133 (red) antibodies (**j**). Magnified boxed regions are shown in the lower left corner. The number of single AL-NPCs in individual AL foci/cluster and the number of cytosolic AL-NPCs per cell were quantified (**k, l**). A total of 677 foci and 6 cells per condition were analyzed (one-way ANOVA; mean ± SD; $N = 3$). Scale bars, 3 µm. Source data are provided as a Source Data file. **m** Asynchronously proliferating 2xZFN-mEGFP-Nup107 HeLa cells treated with the indicated siRNAs for 48 h and co-labeled with anti-p-Rb (blue) and anti-cyclin B1 (red) antibodies. The percentage of cells with small AL foci in p-Rb- and cyclin B1-negative cells (early G1), p-Rb-positive and cyclin B1-negative cells (mid- to late G1 and S phase), and p-Rb- and cyclin B1-positive cells (G2) was quantified; at least 200 cells per condition were analyzed (one-way ANOVA; mean ± SD; $N = 3$). Representative images are shown in Supplementary Fig. 11a. Source data are provided as a Source Data file. **n, o** knockout DLD-1 (**n**) and HCT116 (**o**) cells were treated with DMSO or auxin for 4 h to deplete RanBP2. Cytoplasmic (Cyt) and nuclear (Nuc) fractions were prepared, and Nup levels were analyzed by western blotting (at least 3 independent experiments). Signal intensities were quantified, and cytoplasmic and nuclear signals were normalized to GAPDH and Lamin A, respectively. Mean values are shown below the blots (*$P < 0.05$, **$P < 0.01$, ***$P < 0.001$; paired one-tailed t-test; $N = 3$). **p–s** Representative SMLM immuno-fluorescence images of Nup62 at the nuclear surface in AID RanBP2 knockout DLD-1 cells treated with DMSO or auxin for 4 h (**p**). Magnified boxed regions are shown in the right panels. Nuclear NPC density (Nup62) was quantified (**r**) (unpaired two-tailed t-test; mean ± SD; 50 cells per condition; $N = 3$). Scale bars, 5 µm. Source data are provided as a Source Data file. Representative images of nuclei in AID RanBP2 knockout DLD-1 cells treated with DMSO or auxin for 4 h are shown (**q**), and nuclear area was quantified (**s**). A total of 700 cells per condition were analyzed (unpaired two-tailed t-test; mean ± SD; $N = 3$). Scale bars, 5 µm. Source data are provided as a Source Data file.

the outer and inner rings of AL-NPCs in agreement with reported high-resolution NPC structures[45], and thereby promote AL-NPC assembly in the cytoplasm. Indeed, Nup93 was required for the formation of large AL-foci (Supplementary Fig. 15d-f), and the increase of AL foci abundance by ectopic expression of NT RanBP2 was dependent on Nup93 (Supplementary Fig. 15g, h, j, k, m). Downregulation of Nup93 also led to the formation of Nup62-positive nuclear foci (Supplementary Fig. 15j), which did not co-localize with other Nups, suggesting that they may not represent AL. Co-localization of NT RanBP2 with Nup133 (Supplementary Fig. 15i) and Nup62 (Supplementary Fig. 15l) in the cytoplasm required Nup93, in line with reports showing that Nup93 helps to connect the central channel to the NPC scaffold[52]. We conclude that the function of the NT part of RanBP2 in AL-NPCs can be attributed to its interactions with Nup93 and its ability to stabilize connections between NPC subcomplexes.

Taken together, these results show that RanBP2 drives AL formation through two distinct mechanisms. Under physiological conditions, RanBP2 contributes to scaffold assembly within individual AL-NPCs, including organization of Y-complex components and interactions with Nup93, while under stress conditions it promotes the expansion and clustering of AL-NPCs in the cytoplasm.

## Climp63 directs AL-NPCs to ER sheets and the NE

Our findings on the molecular mechanism of AL-NPC assembly and the direct role of RanBP2 in this process create a unique opportunity to identify additional factors involved in AL biology, in particular those facilitating AL-NPC insertion into the ER membranes. Since the NT RanBP2 fragment can induce the formation of AL (Fig. 6a–g), we analyzed the proteins interacting with HA-tagged NT RanBP2 (aa 1-1171) compared to HA-tag only, using liquid chromatography-mass spectrometry (LC-MS). Of total 97 proteins specifically bound to NT RanBP2, 20 were associated with the NPC, as expected. Additionally, numerous ER-associated proteins were also identified (Supplementary Fig. 16a–c). Among them, Climp63 (also known as CKAP4), a type II membrane protein, emerged as an interesting candidate for the regulation of AL because it has been implicated in maintaining ER morphology, being important for the biogenesis of ER sheets and regulation of their width, and for coordinating the formation and dynamics of ER nanoholes[53,54]. Moreover, its N-terminus binds to MTs, linking the ER to the cytoskeleton[55]. Thus, it is particularly interesting

that we indeed detected an interaction of the HA-tagged NT RanBP2 with endogenous RanBP2 and Climp63 (Fig. 7a, b, Supplementary Fig. 17a). Since stringent cell lysis conditions were used for these experiments and several other ER-resident proteins were not found to interact with RanBP2 (Fig. 7a, Supplementary Fig. 17a), this argues against nonspecific co-purification of bulk ER membranes and supports a more specific association between RanBP2 and Climp63. In contrast to the interaction of RanBP2 with Nups (Fig. 6i), our current data do not allow us to determine whether the association of Climp63 depends on the FG-rich region of the NT part of RanBP2, and mapping the direct Climp63-RanBP2 binding sites therefore represents an important avenue for future work.

In interphase cells, RanBP2-labeled AL foci preferentially localized to ER regions positive for Climp63 (Fig. 7c–e, Supplementary Fig. 17b, c), residing closer to the nucleus relative to the entire ER network (Supplementary Fig. 17d). These Climp63-positive regions likely correspond to ER sheets, in line with published findings[54]. This suggests that, under normal growth conditions, AL may primarily reside within sheet-like regions of the ER rather than in its tubules. The tendency of the ER sheets to localize close to the nucleus may facilitate AL merging with the NE under physiological conditions.

Downregulation of Climp63 (Fig. 7f–k) led to the dispersion of ER sheets throughout the cytoplasm (Fig. 7g), in accordance with previous reports[54], and strongly altered the morphology and distribution of AL foci. AL foci increased in both abundance (Fig. 7g, h) and size (Fig. 7i), relocating to the cell periphery, regions that originally corresponded to ER tubules (Fig. 7g), in Climp63-downregulated cells. Depletion of several ER-shaping and ER-associated proteins including RTN4, ATL1, ATL3, LNPK, KTN1 and P180 did not reproducibly affect AL formation (Supplementary Fig. 17e–g). Instead, their knockdown resulted in highly variable AL phenotypes, likely reflecting general perturbations of ER morphology rather than a specific role in AL biology. Thus, Climp63, and possibly a restricted subset of ER factors, but not ER-associated proteins in general, are specifically involved in the AL pathway. Additionally, the nuclear intensity of RanBP2 was reduced upon Climp63 downregulation (Fig. 7j). SplitSMLM analysis confirmed that Nup foci in Climp63-downregulated cells correspond to AL (Fig. 7k). One can speculate that the reported ability of Climp63 to connect the ER to MTs could explain the observed phenotype and that the absence of Climp63 would inhibit MT-mediated transport of

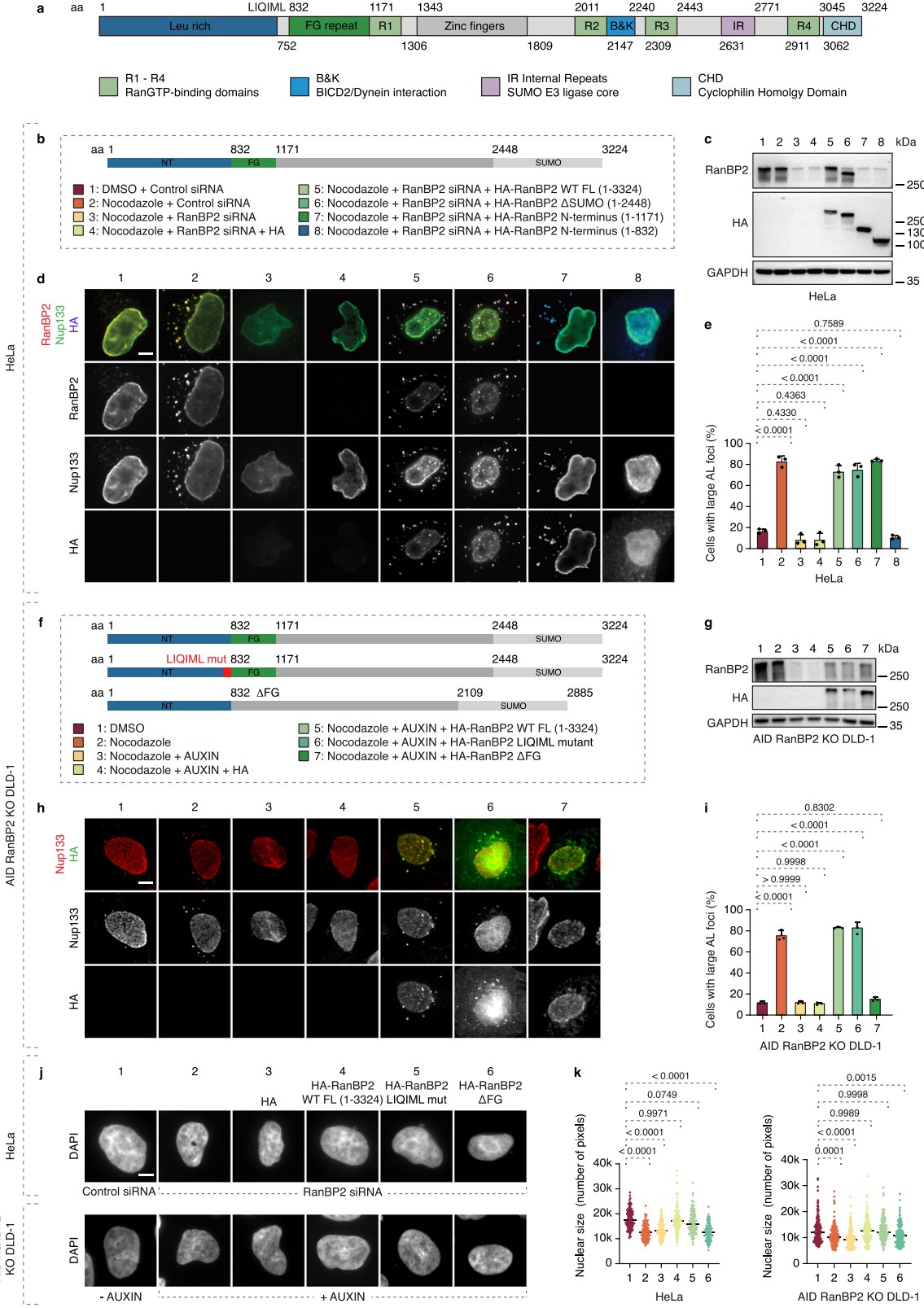

AL towards the NE. Alternatively, Climp63-mediated regulation of ER dynamics and nanomorphology could explain these effects. These observations also suggest that RanBP2-mediated biogenesis of AL-NPCs in the cytosol can be separated from their proper localization at ER membranes, a process directly dependent on Climp63 and that in the absence of Climp63, cells attempt to compensate for these defects by increasing the formation of AL-NPCs.

Irrespective of the underlying mechanisms, which could be a topic for future investigations, downregulation of Climp63 decreased Nup levels at the NE (Fig. 7g, h) and inhibited nuclear expansion (Fig. 7l, m), suggesting that Climp63-mediated ER dynamics is an important prerequisite for the integration of AL-NPCs into the NE in normal cells. Taken together, our results identify the molecular mechanism of AL biogenesis in normal cells, where RanBP2 is crucial for the formation of

**Fig. 5 | RanBP2 FG repeats mediate AL assembly and nuclear function.**
**a** Schematic representation of the domain organization of the RanBP2 protein.
**b–e** Rescue experiments using different RanBP2 protein fragments (**b**). Western blot analysis is shown in (**c**), and representative images of HeLa cells treated with the indicated siRNAs, transfected with different RanBP2 fragments (indicated by numbers in the schematic), and subsequently treated with nocodazole (10 µM) or DMSO for 90 min are shown in (**d**). Cells were co-labeled with anti-Nup133 (green), anti-RanBP2 (red), and anti-HA (blue) antibodies. The percentage of cells containing large AL foci was quantified (**e**); 300 cells per condition were analyzed (one-way ANOVA; mean ± SD; $N = 3$). Scale bars, 5 µm. Source data are provided as a Source Data file. **f–i** Rescue experiments using different full-length (FL) RanBP2 mutant versions (**f**). Western blot analysis of the RanBP2 variants under the indicated conditions is shown in (**g**) (numbers correspond to the schematic). Representative

images of AID-RanBP2 knockout DLD-1 cells treated with DMSO or auxin, transfected with different RanBP2 variants, and subsequently treated with nocodazole (10 µM) or DMSO for 90 min are shown in (**h**). Cells were co-labeled with anti-Nup133 (red) and anti-HA (green) antibodies. The percentage of cells containing large AL foci was quantified (**i**); 300 cells per condition were analyzed (one-way ANOVA; mean ± SD; $N = 3$). Scale bars, 5 µm. Source data are provided as a Source Data file. **j, k** Representative images of nuclei in HeLa cells and AID-RanBP2 knockout DLD-1 cells are shown (**j**). Cells were treated with RanBP2 siRNA or auxin to deplete RanBP2 and subsequently rescued with different RanBP2 variants. A schematic of the RanBP2 variants and western blot analysis are shown in Supplementary Fig. 13j–l. Nuclear size was quantified (**k**); 120 HeLa cells or 150 DLD-1 cells per condition were analyzed (one-way ANOVA; mean ± SD; $N = 3$). Scale bars, 5 µm. Source data are provided as a Source Data file.

---

AL-NPCs in the cytoplasm and Climp63 acts as a factor targeting AL-NPCs to the proper membrane compartment, the ER sheets. These mechanisms ensure nuclear pore assembly and function as well as nuclear growth specifically during the G1 phase of the cell cycle.

### AL-dependent NPC assembly is additive to canonical pathways

What is the relationship of AL-dependent nuclear pore assembly to the canonical assembly pathways acting during two stages of the cell cycle?

Indeed, the postmitotic and the interphase pathways have been primarily described to drive NPC assembly at the NE[26]. Interestingly, in contrast to RanBP2, downregulation of essential upstream components ELYS (postmitotic pathway) or Nup153 (interphase pathway) did not inhibit AL formation but instead strongly induced it (Supplementary Fig. 8a–f)[24]. Our measurements in the G1 cell cycle stage estimated that the contribution of AL to pore assembly is substantial, accounting for 22–45% of total nuclear NPCs, suggesting that somatic cells may utilize an additional assembly pathway governed by RanBP2.

To test this hypothesis, we analyzed the complementarity of the RanBP2-driven AL pathway with the two canonical NPC assembly pathways. Downregulation of ELYS or Nup153 reduced NE-associated Nup levels and NPC density in the nucleus (Fig. 8a–e) in HeLa cells, as expected. Strikingly, additional knockdown of RanBP2 further exacerbated these effects (Fig. 8a–e). The same results were obtained in DLD-1 cells where RanBP2 was depleted for only 4 h using auxin-inducible degradation (Fig. 8f–j). These findings suggest that RanBP2-dependent AL do not act redundantly with the classical NPC assembly pathways but instead contribute independently to the biogenesis of a substantial pool of NPCs at the NE. Taken together, mammalian somatic cells may utilize three distinct nuclear pore assembly pathways, in contrast to the two previously described[26,56–58].

## Discussion

In this study, we demonstrated a universal function and a direct biogenesis mechanism of Annulate Lamellae, mysterious cellular structures known for decades to exist in the cytoplasm[12,13] (Fig. 9). Using splitSMLM[31], an advanced super-resolution microscopy and spectral demixing method for colocalization and image analysis, we show that AL are more abundant than previously recognized and exist in the cytoplasm of a variety of somatic cells under physiological conditions. Our analysis provides unique insights into the biogenesis of AL, which are key cellular organelles involved in the regulation of nuclear function. Compared to healthy state, dysregulation of AL biogenesis occurs in diseased cells, where AL accumulate and cluster in the cytoplasm, as exemplified by our data in the fragile X syndrome patient-derived samples (Fig. 1d). Under physiological conditions during the G1 cell cycle stage, AL merge with the NE and supply it with pre-formed NPCs, a process that is particularly active during periods of rapid nuclear growth (Fig. 9). Inhibition of AL merging with the NE, either by blocking their MT-based movement (Fig. 3d–f), interfering with AL

biogenesis through RanBP2 deletion (Fig. 4n–s), or by preventing their Climp63-mediated localization to ER membranes (Fig. 7f–m), reduces the density of nuclear pores and inhibits nuclear expansion during G1. These findings identify a specific physiological context that drives the use of AL in normal cells, which were previously not amenable to AL analysis. Our data may also explain why, in the past, AL have been commonly observed in rapidly developing, cancerous, or challenged cells, which are likely to depend more crucially on rapid nuclear growth[3–11]. Under these conditions, cells might actively increase their cytoplasmic AL pool by upregulating RanBP2-dependent mechanisms. In addition, our findings show that AL vary widely in size and abundance and are highly dynamic undergoing continuous fusion and fission prior to the merging with the NE (Fig. 2d, e). This dynamic behavior may further explain why previous static analyses failed to detect AL in somatic cells. Given the variable abundance of AL in different cell types (Fig. 1f), we cannot exclude the possibility that they may rely on the AL pathway to varying degrees.

Blocks of AL structures inserted into the NE have been previously observed in *Drosophila* embryos[7], while our study demonstrates that this process is evolutionarily conserved and operates in somatic cells in general. AL have also been implicated in merging with the NE in postmitotic cells[15], however, in contrast to our work, this phenomenon seemed independent of ER-NE contact sites, and rather involved direct interactions with decondensing chromosomes during late anaphase. Time-resolved analysis of AL dynamics allowed us to estimate the net contribution of AL-NPCs to the NE, which accounts for more than 900 new pores during a single G1 phase of the cell cycle, representing a substantial portion, approximately one-third, of the total present NE-NPCs (Fig. 2d, e). Analysis of the NPC assembly kinetics revealed that postmitotic NPC assembly is completed within 15 min, rapidly adding around two thousand NPCs, while by contrast, interphase NPC assembly is slower and more sporadic, plateauing after 100 minutes, with an unknown NPC assembly rate[59]. In comparison, our AL-NPC insertion data show a continuous rate of 1.5 NPCs per minute throughout the entire G1, as supported by the persistent presence of AL foci from early G1 through mid-to-late G1/S (Fig. 2a, b) and by photoconversion experiments (Fig. 2f, g). Thus, although slower than postmitotic assembly, AL-NPC insertion occurs at a rate comparable to canonical interphase assembly and represents a kinetically significant and sustained mechanism of NPC addition.

Strikingly, AL-driven NPC assembly acts complementarily to the two canonical NPC assembly pathways, and the effects on pore density observed upon simultaneous depletion of RanBP2 and ELYS (postmitotic pathway) or Nup153 (interphase pathway) are additive (Fig. 8a–j). Thus, RanBP2- and AL-dependent NPC addition occurs through an independent mechanism (Fig. 8a–j). Supporting this, our data identify a RanBP2 mutant (ΔFG region mutant) that clearly separates its AL-specific function from its other roles at the NE and in the cytoplasm. Taken together, our study positions the AL-insertion mechanism as a distinct, possibly third NE-NPC biogenesis pathway

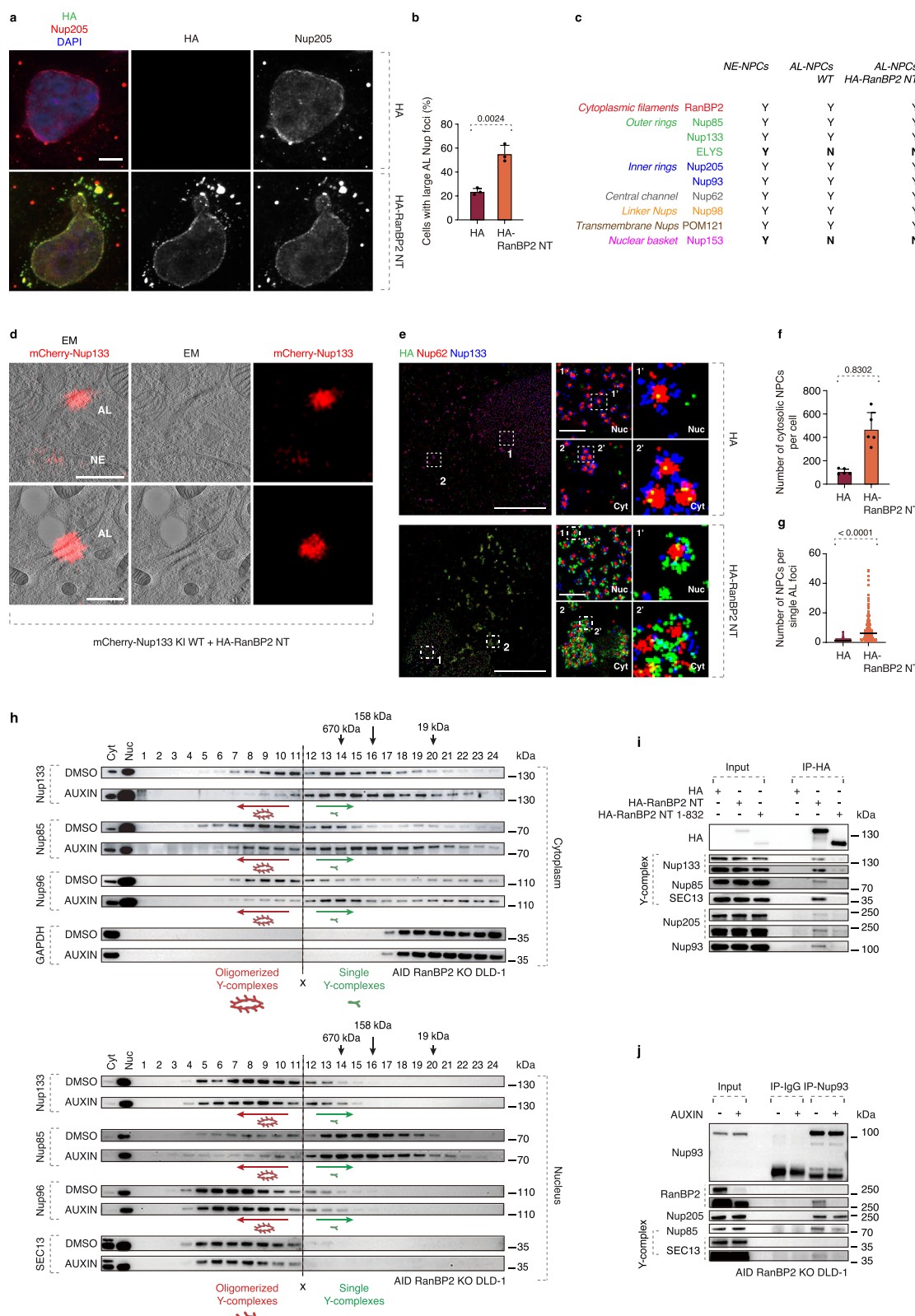

that operates specifically during early interphase and acts complementarily to the two well-described pathways in higher eukaryotes[26,56–58]. In the future, it will be important to understand why downregulation of factors responsible for postmitotic and interphase NPC assembly results in the accumulation of AL and how these defects are linked to the regulation of RanBP2.

Our findings on the existence of small AL structures represent an important first step toward future studies aimed at understanding how normal somatic cells acquire pre-assembled AL-NPCs during nuclear growth. During development, the simultaneous insertion of numerous AL-NPCs (arranged within several parallel ER membranes in the cytoplasm) into the NE involves the formation of large NE openings[7], which

**Fig. 6 | RanBP2 drives oligomerization of AL-NPCs. a**, **b** Representative images of HeLa cells transfected with HA or an HA-RanBP2 N-terminal (NT) fragment. Cells were co-labeled with DAPI (blue), anti-Nup205 (red), and anti-HA (green) antibodies (**a**). The percentage of cells containing large AL foci was quantified (**b**); at least 300 cells per condition were analyzed (unpaired two-tailed t-test; mean ± SD; $N = 3$). Scale bar, 5 μm. Source data are provided as a Source Data file. **c** Summary of nucleoporin composition in NE-NPCs, WT ALC-NPCs, and AL-NPCs induced by RanBP2-NT expression. Colors indicate NPC subcomplexes; Y, present; N, absent. Representative images are shown in Supplementary Fig. 14a. **d** Representative correlative light and electron microscopy (CLEM) images of mCherry-Nup133 knock-in (KI) HeLa cells transfected with the HA-RanBP2-NT fragment ($N = 3$). Magnified boxed regions are shown in the corresponding numbered panels. mCherry fluorescence accumulates at densely packed AL-NPCs on stacked ER sheets in the cytoplasm, corresponding to annulate lamellae (AL), and is also observed at the nuclear envelope (NE). Scale bars, 1 μm. **e**–**g** Representative splitSMLM images showing NPCs at the nuclear surface (Nuc) and AL-NPCs in the cytoplasm (Cyt) of HeLa cells transfected with HA or HA-RanBP2-NT. Nup133 labels the cytoplasmic and nuclear rings of NPCs, and Nup62 marks the central channel. Magnified boxed regions are shown in the corresponding numbered (**e**). The total number of cytosolic AL-NPC complexes per cell and the number of individual AL-

NPC complexes in a single AL foci/cluster were quantified (**f**, **g**). 6 cells per condition were analyzed (two-tailed unpaired t-test; mean ± SD; $N = 3$). Scale bars, 3 μm (whole cell) and 0.3 μm (zoom). Source data are provided as a Source Data file. **h** Auxin-inducible degron (AID) RanBP2 knockout DLD-1 cells were treated with DMSO or auxin for 4 h to deplete RanBP2. Cytoplasmic and nuclear fractions were prepared, separated by Superose 6 size-exclusion chromatography, and analyzed by western blotting (at least 3 independent experiments). Fraction 1 corresponds to the void volume; elution positions of thyroglobulin (670 kDa), γ-globulin (158 kDa), and myoglobin (19 kDa) are indicated. Green arrows indicate fractions enriched in single Y-complexes, and red arrows indicate higher-molecular-weight fractions containing oligomerized Y-complexes. The vertical dashed line and X symbol indicate the boundary between separately run gels. Depletion of RanBP2 reduced the abundance of oligomerized Y-complexes in cytoplasmic but not nuclear fractions. **i** Lysates of HeLa cells expressing HA, HA-RanBP2-NT, or HA-RanBP2-NT ΔFG were immunoprecipitated with anti-HA magnetic beads and analyzed by western blotting (at least 3 independent experiments). **j** AID RanBP2 knockout DLD-1 cells were treated with auxin for 4 h to deplete RanBP2. Cell lysates were immunoprecipitated with an anti-Nup93 antibody or IgG control and analyzed by western blotting (at least 3 independent experiments).

---

likely serve to avoid topological constraints associated with AL insertion. It is unknown if a similar mechanism operates in somatic cells. Studies by Otsuka and Ellenberg[60] indicate that the reforming NE is fenestrated at mitotic exit and largely seals by early G1; however, a small fraction of openings may plausibly persist into G1. Such residual discontinuities could, in principle, serve as entry sites for AL-derived NPCs or for membrane fusion events contributing to NE growth, which should be studied in the future. Our splitSMLM analysis reveals that small, symmetric AL-NPCs laterally merge with asymmetric NE-NPCs. Some of these fusion events are accompanied by the formation of small NE openings containing ER membranes, with both AL-NPCs and NE-NPCs positioned adjacent to each other, without disrupting the underlying lamina components (Fig. 3a–c). While a recently proposed model suggests that the GTPase Ran remodels AL-NPCs into asymmetric, functional NPCs in fly embryos[34], it will be crucial in future work to define the exact remodeling steps that occur during AL membrane insertion into the NE in somatic cells, ideally using high-resolution ultrastructural approaches. Although our data support the insertion of intact AL into the NE, we cannot yet rule out the possibility that AL-NPCs undergo partial disassembly prior to insertion, or that AL act primarily as donors of NPC subcomplexes rather than inserting as complete structures. Interestingly, the bleb-like AL-NE intermediates observed on the NE by the splitSMLM analysis may resemble NE herniations described in *Saccharomyces cerevisiae* and in human cells carrying mutations in Brl1/6 or TorsinA[61–65]. These factors have been implicated in interphase NPC assembly, possibly through roles in membrane fusion and lipid homeostasis at the NE. Whether Brl1/6 or TorsinA also contribute to AL insertion into the NE and the regulation of AL-NPCs remains an important avenue for future research. Yet, our work not only demonstrates that small AL are clearly beneficial for cellular homeostasis by preserving nuclear function during G1 but also lays a solid foundation for important future studies on the molecular topology of nuclear pore insertion into the NE.

In addition to elucidating the universal cellular function of AL during G1, the direct molecular mechanisms of AL formation identified in our study explain how AL-NPCs are made and how they could localize to the proper compartment of the ER (Fig. 9). We show that RanBP2 is indispensable for the assembly of cytosolic AL-NPCs across cell types and conditions, serving as an assembly platform that specifically regulates the oligomerization of Y complexes within the AL-NPC scaffold. While this process may involve other structurally important Nups, such as Nup93 (Supplementary Fig. 15), the N-terminal region of RanBP2 containing FG repeats is required for AL biogenesis. Intrinsically disordered FG repeats have been shown to stabilize

subcomplexes within NE-NPCs[51], providing a possible explanation for their function in AL-NPCs. The role of cohesive FG repeats could also help explain previous observations in fly embryos, where RanBP2 and intact microtubules were implicated in recruiting various condensed precursor Nup particles, predicted to lead to AL biogenesis[10]. Although we occasionally observe cytoplasmic RanBP2 foci under normal conditions (Fig. 3c), we are unable to clearly classify them as membrane-free. Interestingly, our results reveal a spatial association of AL with the microtubule network (Supplementary Fig. 7b) and RanBP2 has previously been reported to stabilize MT bundles and recruit dynein motor proteins[66,67]. Whether RanBP2 mediates the interaction between AL and microtubules and if this process involves the formation of Nup condensate particles that facilitate AL assembly in somatic cells, remain important questions for the future. Here, we provide direct evidence for the role of RanBP2 in AL biogenesis and clarify the detailed molecular requirements for AL-NPC formation that is functionally analogous but mechanistically distinct from that described in fly oocytes.

On the membrane side, we identify Climp63 as a key factor localizing AL-NPCs to ER sheets proximal to the nucleus, a process important for their subsequent integration into the NE and nuclear expansion. To our knowledge, this is the first evidence directly linking AL to a specific ER substructure so far. By analyzing the interaction and function of several other ER factors, we support the idea that Climp63, and possibly a restricted subset of ER proteins, but not ER-associated factors in general, is specifically involved in the AL pathway. Since Climp63 was shown to ensure not only the formation of ER sheets but also the biogenesis of ER nanoholes[53,54], which were suggested to be similar in size to that of nuclear pores[53], it would be exciting to study the roles of ER dynamics and nanomorphology in AL biogenesis. It would also be important to understand if the reported role of Climp63 in connecting the ER to MTs could explain its role in positioning AL in proximity to the NE.

One of the most abundant proteins in the AL proteome belongs to the RGPD family (Supplementary Fig. 16a), which shares significant sequence similarity with the N-terminal domain of RanBP2. Although RGPD proteins are predicted to localize to the Golgi apparatus, they may retain MT-binding capacity[68] and it will be interesting to study this protein family in the context of AL biology. Likewise, a possible role of NE-resident proteins SUN1 and SUN2, which we found to co-localize with AL foci (Supplementary Fig. 5b) and which are components of LINC complexes[69], could be relevant during the early steps of NPC interphase assembly[70], which opens possibilities for future investigations.

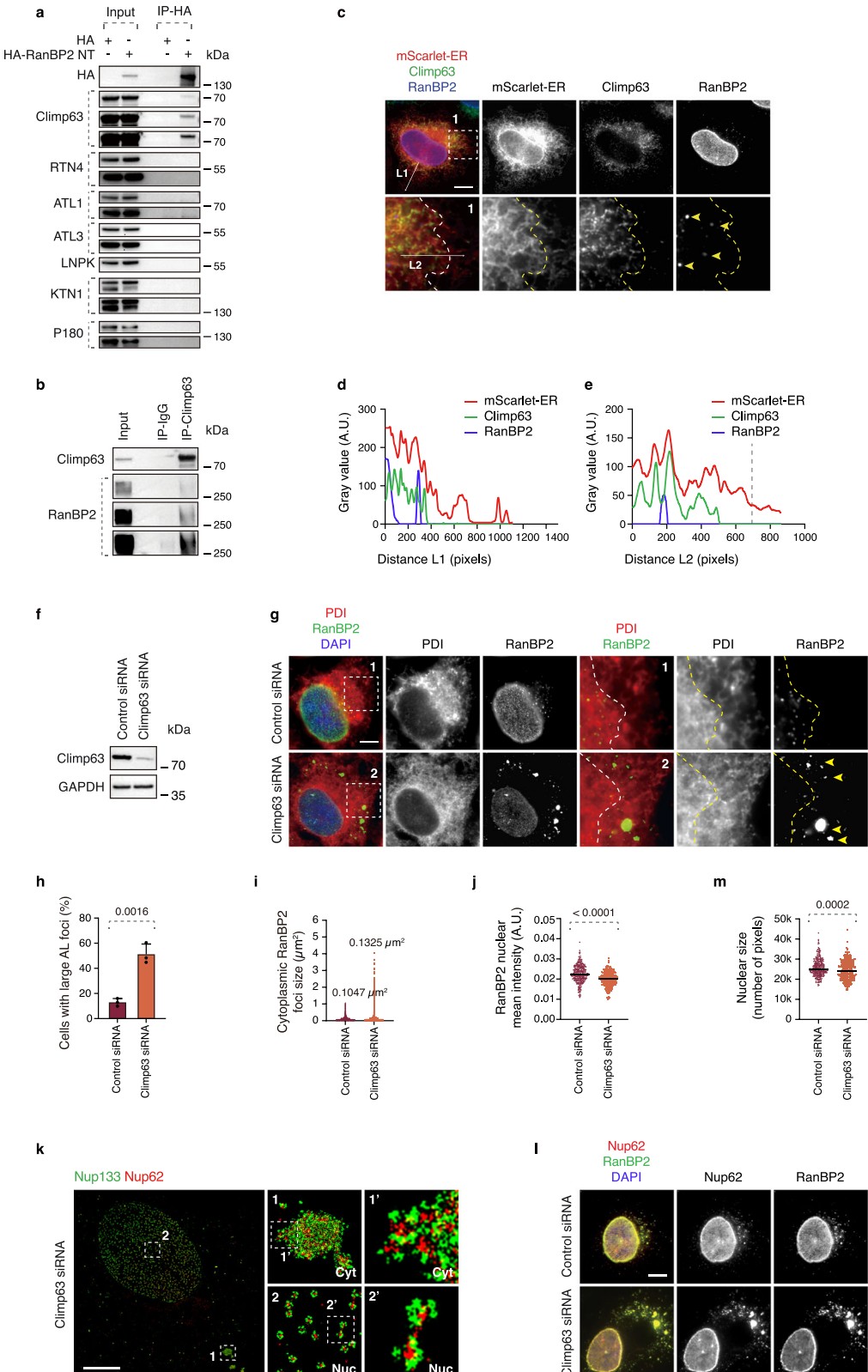

Undoubtedly, our findings not only clarify the fundamental mechanisms of AL biogenesis but also lay an important foundation for future investigations aiming to identify AL-specific factors, which could be utilized by pathologically challenged cells such as during malignancy. Indeed, maintaining NPC function is crucial for normal cellular physiology, as disruptions in nucleocytoplasmic transport,

mislocalization of Nups, and elevated levels of AL have been observed in various diseased and infected cells[23,25,46,71–76]. The biological principles of AL biology identified by our study may create a basis for future approaches to restore NPC homeostasis in human diseases such as fragile X syndrome and numerous other neurological and neurodegenerative diseases as well as cancer[14].

**Fig. 7 | RanBP2 interacts with Climp63 that localizes AL-NPCs to ER sheets.**
**a** Lysates of HeLa cells expressing HA or HA-RanBP2 N-terminal (NT) were immu-noprecipitated with anti-HA magnetic beads and analyzed by western blotting (at least 3 independent experiments). **b** Lysates of HeLa cells were immunoprecipitated with an anti-Climp63 antibody or IgG control and analyzed by western blotting (at least 3 independent experiments). **c**–**e** Representative images of HeLa cells expressing the ER marker mScarlet-ER. Cells were co-labeled with anti-Climp63 (green) and anti-RanBP2 (blue) antibodies (**c**). Magnified boxed regions are shown in the corresponding numbered panels. The dotted line indicates the boundary between Climp63-labeled ER sheets and more distal ER tubules. Intensity profiles along arrows L1 and L2 showing signals from all three channels are shown (**d**, **e**). **f**–**j** HeLa cells treated with the indicated siRNAs were analyzed by western blotting (**f**). Cells were co-labeled with anti-RanBP2 (green) and anti-PDI (red) antibodies (**g**). Magnified boxed regions are shown in the corresponding numbered panels. The dotted line delineates cytoplasmic regions containing ER sheets proximal to the nucleus, where AL-NPC foci are enriched in WT cells. Upon Climp63 depletion, AL-NPC foci enlarge and localize more distally. The percentage of cells with large AL foci was quantified (**h**); at least 400 cells per condition were analyzed (unpaired two-tailed t-test, mean ± SD, $N = 3$). The size of cytoplasmic AL foci was quantified (**i**; at least 3500 foci per condition were analyzed ($N = 3$). Nuclear envelope RanBP2 intensity was quantified (**j**), where at least 280 cells per condition were analyzed (unpaired two-tailed t-test, mean ± SD, $N = 3$). Scale bars, 5 μm. **k** Representative splitSMLM images depicting NPCs on the nuclear (Nuc) surface and AL-NPCs in the cytoplasm (Cyt) of HeLa cells treated with Climp63 siRNA ($N = 3$). Nup133 labels the cytoplasmic and nuclear rings of the NPC, and Nup62 marks the central channel. Magnified boxed regions are shown in the corresponding numbered panels. Scale bars, 3 μm. **l**, **m** Representative images of HeLa cells treated with the indicated siRNAs. Cells were co-labeled with anti-RanBP2 (green), anti-Nup62 (red), and DAPI (**l**). Nuclear size was quantified (**m**), and at least 300 cells per condition were analyzed (unpaired two-tailed t-test, mean ± SD, $N = 3$). Scale bars, 5 μm.

## Methods
### Antibodies
The following primary antibodies were used: rabbit monoclonal anti-Nup133 (Abcam, ab155990, 1:1000 for WB, 1:500 for IF), mouse monoclonal anti-Nup133 (E-6, Santa Cruz Biotechnology, sc-376763, 1:1000 for WB, 1:500 for IF), mouse monoclonal anti-Nucleoporin p62 (BD Biosciences, 610497,1:1000 for WB, 1:200 for IF), rabbit polyclonal anti-RanBP2 (Abcam, ab64276, 1:1000 for WB, 1:1000 for IF), rabbit polyclonal anti-Nup214 (Abcam, ab70497, 1:1000 for WB, 1:500 for IF), rabbit polyclonal anti-Nup85 (Bethyl, A303-977A, 1:1000 for WB, 1:500 for IF), mouse monoclonal anti-Pericentrin 1 (D-4, Santa Cruz Biotechnology, sc-376111, 1:1000 for WB, 1:500 for IF), rabbit monoclonal anti-SEC13 (R&D Systems, MAB9055, 1:1000 for WB, 1:500 for IF), rabbit polyclonal anti- ELYS (Bethyl, A300-166A, 1:500 for WB, 1:500 for IF), mouse monoclonal Nup205 (H-1, Santa Cruz Biotechnology, sc-377047, 1:1000 for WB, 1:500 for IF), mouse monoclonal anti-Nup93 (E-8, Santa Cruz Biotechnology, sc-374399, 1:1000 for WB, 1:500 for IF), rabbit monoclonal anti-Nup98 (C39A3, Cell Signaling Technology, 2598, 1:1000 for WB, 1:500 for IF), rabbit polyclonal anti-Nup96 (Bethyl, A301-784A, 1:1000 for WB, 1:500 for IF), rabbit polyclonal anti-POM121 (GeneTex, GTX102128, 1:1000 for WB, 1:250 for IF), rabbit polyclonal anti-Nup153 (Abcam, ab84872, 1:1000 for WB, 1:500 for IF), rabbit polyclonal anti-CRM1/Exportin 1 (Novus Biologicals, NB100-79802, 1:500 for IF), mouse monoclonal anti-NTF97/Importin beta (Abcam, ab2811, 1:500 for IF), mouse monoclonal anti-Ran (BD Biosciences, 610340, 1:1000 for WB, 1:500 for IF), rabbit monoclonal anti-SUN1 (Abcam, ab124770, 1:500 for IF), rabbit monoclonal anti-SUN2 (Abcam, ab124916, 1:500 for IF), rabbit polyclonal anti-Lamin A (C-terminal) (Sigma, L1293, 1:1000 for WB, 1:500 for IF), rabbit polyclonal anti-Lamin B1 (Abcam, ab16048, 1:1000 for WB, 1:500 for IF), rabbit monoclonal anti-Nesprin1 (Abcam, ab192234, 1:500 for IF), rabbit polyclonal anti-Emerin (Abcam, ab40688, 1:500 for IF), rabbit polyclonal anti-LAP2 (Proteintech, 14651-1-AP, 1:500 for IF), rabbit polyclonal anti-Phospho-Rb (Ser807/811, Cell Signaling Technology, 9308, 1:500 for IF), mouse monoclonal anti-Cyclin B1 (G-11, Santa Cruz Biotechnology, sc-166757, 1:1000 for WB, 1:500 for IF), rabbit polyclonal anti-Cyclin B1 (GeneTex, GTX100911, 1:1000 for WB, 1:500 for IF), rat monoclonal anti-HA (Roche, 11867423001, 1:500 for WB, 1:500 for IF), rabbit polyclonal anti-GAPDH (Sigma, G9545, 1:10,000 for WB), mouse monoclonal anti-β-Actin (Sigma, A2228, 1:10000 for WB), mouse monoclonal anti-mCherry (4B3, Thermo Scientific, MA5-32977, 1:1000 for WB, 1:500 for IF), rabbit polyclonal anti-UBAP2L (Abcam, ab138309, 1:1000 for WB, 1:500 for IF), mouse monoclonal anti-FXR1 (Millipore, 03-176, 1:1000 for WB, 1:500 for IF), rabbit polyclonal anti-α-tubulin (Abcam, ab18251, 1:10,000 for WB, 1:500 for IF), mouse monoclonal anti-α-tubulin (Sigma, T9026, 1:5000 for WB, 1:500 for IF), mouse monoclonal anti-Nuclear Pore Complex Proteins (mAb414, Abcam, ab24609, 1:1000 for WB, 1:500 for IF), rabbit polyclonal anti-LC3B (Novus Biologicals, NB100-2220SS, 1:1000 for WB), rabbit polyclonal anti-LC3A (Novus Biologicals, NB100-2331, 1:1000 for WB), rabbit polyclonal anti-p62 (GeneTex, GTX100685, 1:1000 for WB, 1:500 for IF), rat monoclonal anti-GFP (3H9, ChromoTek, 3h9-100, 1:1000 for WB, 1:500 for IF), rabbit polyclonal anti-GFP (Abcam, ab290, 1:1000 for WB, 1:500 for IF), rabbit polyclonal anti-Aurora B (Abcam, ab2254, 1:500 for IF), rabbit polyclonal anti-cyclin A (H-432, Santa Cruz Biotechnology, sc-751, 1:1000 for WB, 1:500 for IF), mouse monoclonal anti-PDI (Abcam, ab2792, 1:500 for IF), mouse monoclonal anti-Climp63 (Enzo, ALX-804-604, 1:1000 for WB, 1:500 for IF), rabbit polyclonal anti-CKAP4 (Proteintech, 16686-1-AP, 1:1000 for WB, 1:500 for IF), rat monoclonal α-tubulin-conjugated to Alexa Fluor® 647 (Abcam, ab195884, 1:500 for IF), rabbit polyclonal anti-RTN4/NOGO (Proteintech, 10950-1-AP, 1:1000 for WB), rabbit polyclonal anti-RRBP1 (Proteintech, 22015-1-AP, 1:1000 for WB), rabbit polyclonal anti- KTN1 (Proteintech, 19841-1-AP, 1:1000 for WB), rabbit polyclonal anti-ATL1(OriGene, ta332595, 1:1000 for WB), rabbit polyclonal anti-ATL3 (Proteintech, 16921-1-AP, 1:1000 for WB), rabbit polyclonal anti-LNPK (Atlas Antibodies, HPA014205, 1:1000 for WB).

Secondary antibodies used were the following (HRP-conjugated antibodies were used at a dilution of 1:5000, while fluorescent-labeled antibodies were used at a dilution of 1:500): goat anti-mouse IgG antibody (HRP) (GeneTex, GTX213111-01), goat anti-rabbit IgG antibody (HRP) (GeneTex, GTX213110-01) and goat anti-rat IgG antibody (HRP) (Cell Signaling Technology, 7077S), goat polyclonal anti-mouse AF647 (Thermo Fisher Scientific, A-21236), goat polyclonal anti-mouse AF568 (Thermo Fisher Scientific, A-11031), goat polyclonal anti-mouse AF555 (Thermo Fisher Scientific, A-21424), goat polyclonal anti-mouse AF488 (Thermo Fisher Scientific, A-11029), goat polyclonal anti-rabbit AF647 (Thermo Fisher Scientific, A-21245), goat polyclonal anti-rabbit AF568 (Thermo Fisher Scientific, A-11036), goat polyclonal anti-rabbit AF555 (Thermo Fisher Scientific, A-21429), goat polyclonal anti-rabbit AF488 (Thermo Fisher Scientific, A-11034), goat polyclonal anti-mouse CF680 (Sigma, SAB4600199), goat polyclonal anti-chicken CF660C (Sigma, SAB4600458), goat polyclonal anti-rat CF660C (Sigma, SAB4600193).

### Plasmid and siRNA transfections
The following plasmids were used: pX330-P2A-EGFP/RFP and pUC57 was a generous gift from Zhirong Zhang (Romeo Ricci laboratory IGBMC). pQCXIP-mScarlet-ER was a generous gift from Julie Eichler[77] (Catherine-Laure Tomasetto laboratory IGBMC). pEF-HA-MCS2, pEF-HA-RanBP2 (1-3224), pEF-HA-RanBP2 (1-2448), and pEF-HA-RanBP2 (1-1171) were generous gifts from Ralph Kehlenbach[78] (University of Göttingen). pEF-HA-RanBP2 (1-832) and pEF-HA-RanBP2 (FL without FG repeat) was obtained by inserting the target sequence, obtained by PCR using pEF-HA-RanBP2 as a template, into pEF-HA-MCS2. pCDNA 3.1 3XHA NUP358 FL (LIQIML) was generous gifts from André Hoelz

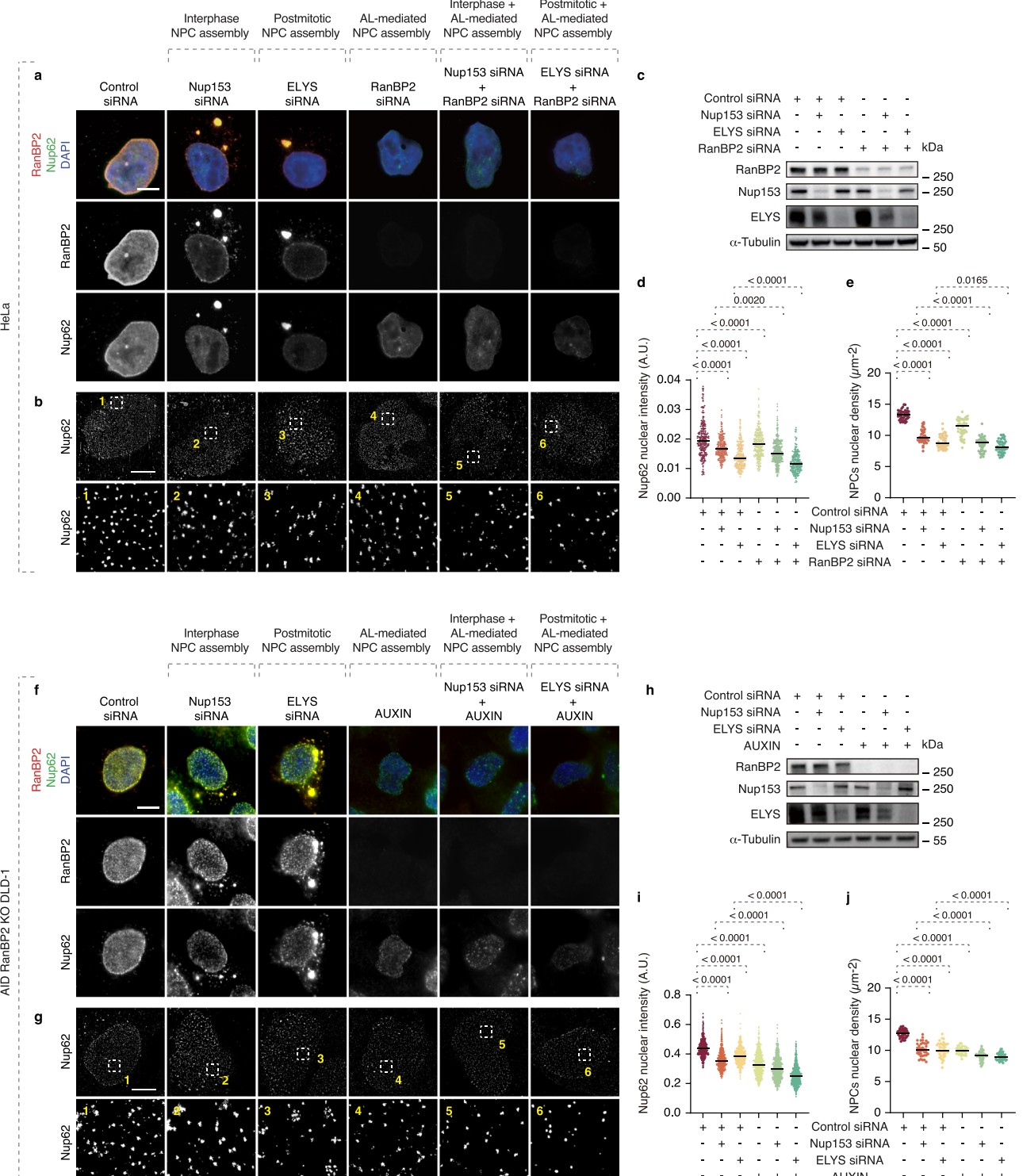

**Fig. 8 | AL-dependent NPC assembly is additive to canonical pathways.**
**a**–**e** Representative images of HeLa cells treated with the indicated siRNAs for 48 h. Cells were co-labeled with anti-Nup62 (green) and anti-RanBP2 (red) antibodies for immunofluorescence microscopy (**a**) or labeled with anti-Nup62 for SMLM immunofluorescence (**b**). Samples from the indicated conditions were analyzed by western blotting (**c**). Nuclear envelope (NE) Nup62 intensity was quantified (**d**); 200 cells per condition were analyzed (one-way ANOVA test, mean ± SD, N = 3). Scale bars, 5 μm. Nuclear NPC density (Nup62) was quantified (**e**); 50 cells per cell line were analyzed (one-way ANOVA test, mean ± SD, N = 3). Scale bars, IF, 5 μm; SMLM, 3 μm. Source data are provided as a Source Data file. **f**–**j** Representative images of

auxin-inducible degron (AID) RanBP2 knockout DLD-1 cells treated with auxin or the indicated siRNAs for 48 h. Cells were co-labeled with anti-Nup62 (green) and anti-RanBP2 (red) antibodies for immunofluorescence microscopy (**f**) or labeled with anti-Nup62 for SMLM immunofluorescence (**g**). Samples from the indicated conditions were analyzed by western blotting (**h**). Nuclear envelope Nup62 intensity was quantified (**i**); 200 cells per condition were analyzed (one-way ANOVA test, mean ± SD, N = 3). Scale bars, 5 μm. Nuclear NPC density (Nup62) was quantified (**j**); 50 cells per cell line were analyzed (one-way ANOVA test, mean ± SD, N = 3). Scale bars, IF, 5 μm; SMLM, 3 μm. Source data are provided as a Source Data file.

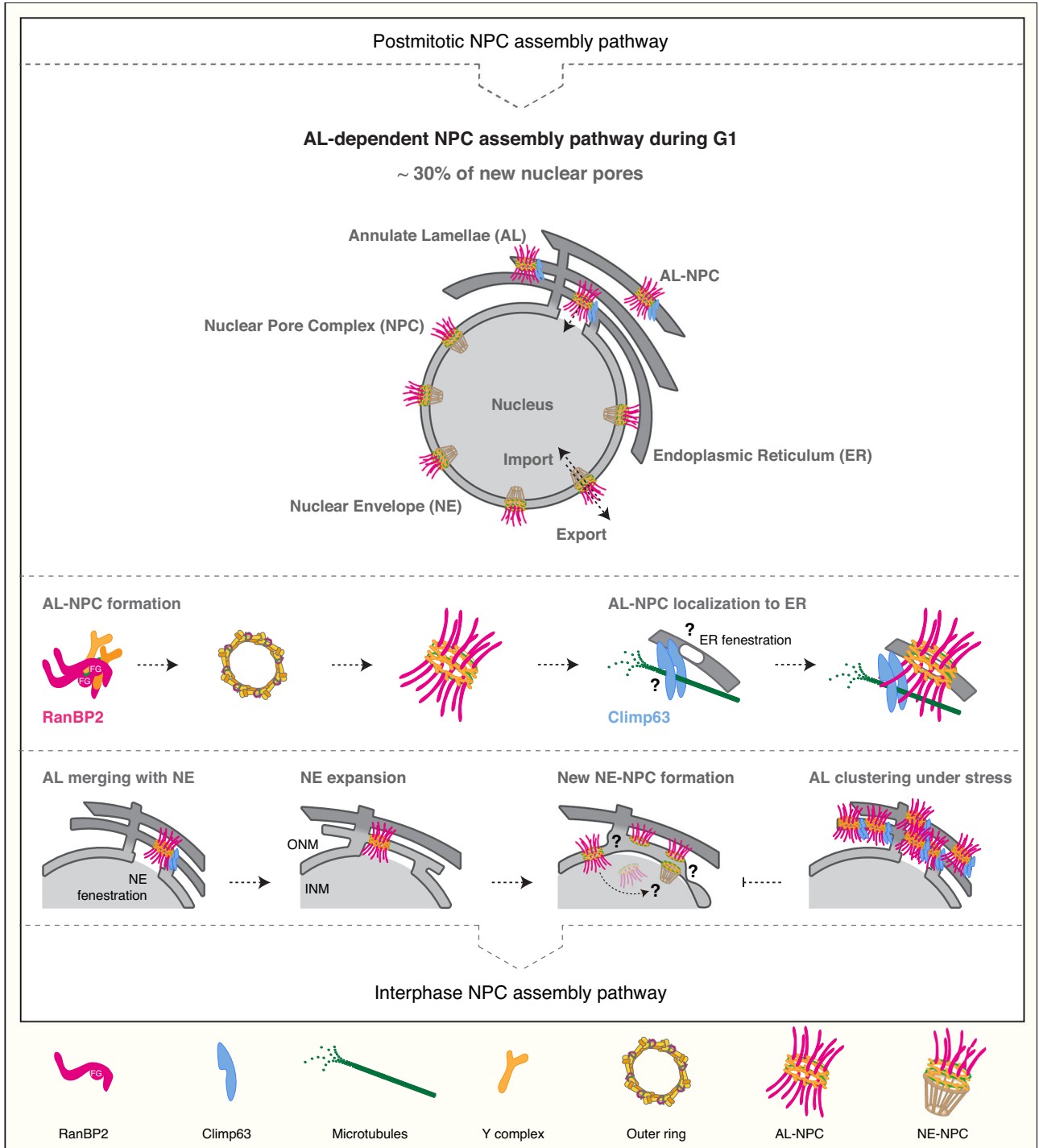

**Fig. 9 | Model of an AL-dependent NPC assembly pathway.** A hypothetical model depicting the universal function and direct biogenesis mechanisms of annulate lamellae (AL). The scheme illustrates a distinct NPC biogenesis route mediated by AL, mechanistically separate from both postmitotic and interphase NPC assembly pathways. AL containing a limited number of AL-NPCs embedded in ER-derived membrane sheets are present in the cytoplasm of normally proliferating somatic cells and play a crucial role in supplying the nuclear envelope (NE) with new nuclear pores. During G1, this AL-dependent pathway contributes to nuclear growth by delivering preformed NPCs to the NE, accounting for an estimated ~30% of newly incorporated nuclear pores and supporting increased nucleocytoplasmic transport during nuclear expansion. AL-NPC biogenesis is initiated in the cytoplasm and requires RanBP2 (pink). The N-terminal region of RanBP2, including its phenylalanine-glycine (FG) repeats, promotes oligomerization of NPC scaffold components (Y-complexes, orange) outside the NE, enabling AL-NPC pre-assembly. The ER-resident protein Climp63 (blue) subsequently mediates the targeting and insertion of AL-NPCs into ER sheets positioned near the NE, thereby spatially coupling AL-NPCs to the nuclear periphery. During G1, AL membranes merge with the expanding NE, likely via transient NE fenestrations, allowing AL-NPCs to be incorporated into the nuclear envelope and potentially remodeled into mature NE-NPCs. This AL-driven mechanism operates in parallel to, but independently of, canonical postmitotic and interphase NPC assembly, defining a third NPC assembly pathway. Under pathological or stress conditions, AL fail to merge with the NE and instead accumulate and cluster within the cytoplasm, preventing AL-NPC transfer to the nucleus. Collectively, this model defines AL as a dynamic cytoplasmic NPC reservoir with a dedicated biogenesis mechanism and a crucial role in nuclear pore homeostasis.

(California Institute of Technology). The siRNA-resistant RanBP2 plasmid was obtained by PCR and DpnI (Thermo) treatment. NLS-mCherry-LEXY (pDN122) (Plasmid #72655) and mEos2-N1 (Plasmid #54662) was purchased from Addgene. The primers used for plasmid generation are provided in the Supplementary Table 1.

The following siRNA oligonucleotides were used: Non-targeting individual siRNA (Dharmacon), FXR1 siRNA (Dharmacon), UBAP2L siRNA (Dharmacon), RanBP2 siRNA (Dharmacon), Nup153 siRNA (Dharmacon), ELYS siRNA (Dharmacon), POM121 siRNA (Life Technologies), Nup214 siRNA (Eurogenetec), Nup93 siRNA (Horizon), CLIMP-63(Dharmacon), RTN4 (Horizon), p180/RRBP1 (Horizon), KTN1 (Horizon), ATL1 (Horizon), ATL3 (Horizon), LNPK (Horizon). The sequences of siRNA are provided in the Supplementary Table 1.

X-tremeGENE™ 9 DNA Transfection Reagent (Roche) was used to perform transient plasmid transfections according to the manufacturer's instructions. Lipofectamine™ RNAiMAX Transfection Reagent (Invitrogen) was used to deliver siRNAs for gene knock-down (KD) according to the manufacturer's instructions.

### Generation of cell lines and cell culture

HeLa Kyoto (human cervix carcinoma) cells and U2OS (human bone osteosarcoma) cells were purchased from ATCC. HeLa Kyoto GFP-Nup107 derived stable cell were purchased from CSL cell bank. AID-NUP358 DLD-1 (colorectal adenocarcinoma epithelial cells) cells and AID-NUP358 HCT116 (human colon cancer cells) cells were generous gifts from Mary Dasso[42,45] (National Institutes of Health). hTERT-RPE1 (Human retinal pigment epithelial-1) cells was a generous gift from Juliette Godin (IGBMC). FB789 (Human Fibroblast) cells was a generous gift from Luca Proietti (IGBMC). MRC5 (Human Fibroblast) cells was a generous gift from Thierry Seroz (IGBMC).Control human fibroblasts and FXS patient-derived fibroblasts were previously reported[25].

mCherry-Nup133 and mEos2-Nup133 knock-in HeLa Kyoto cell lines were generated using CRISPR/Cas9 genome editing system as described previously[79]. The guide RNAs (gRNA) were designed as described previously[80], and cloned into pX330-P2A-EGFP through ligation using T4 ligase (New England Biolabs). The donor constructs used as templates for homologous recombination to repair the Cas9-induced double-strand DNA breaks were generated by cloning the Nup133 DNA fragments upstream and downstream of the CRISPR target sequences and the sequence for mCherry or mEos2, assembled using ExonucleaseIII (Takara) method, into a pUC57 vector. HeLa Kyoto cells were transfected for 24 h and GFP positive cells were collected by FACS (BD FACS Aria II), cultured for 2 days and seeded with FACS into 96-well plates. Single-cell clones of mCherry-Nup133 knock-in or mEos2-Nup133 knock-in were validated by Western blot and sequencing of PCR-amplified targeted fragment by Sanger sequencing (GATC). The sequences of gRNA and the primers used for plasmid generation are provided in the Supplementary Table 1.

UBAP2L knock-out (KO) in mCherry-Nup133 knock-in HeLa Kyoto cell lines were generated using CRISPR/Cas9 genome editing system as described previously[79]. Two guide RNAs (gRNA) were designed using the online software Benchling (https://www.benchling.com/), and cloned into pX330-P2A-EGFP/RFP through ligation using T4 ligase (New England Biolabs). mCherry-Nup133 knock-in HeLa Kyoto cells were transfected for 24 h and GFP and RFP double positive cells were collected by FACS (BD FACS Aria II), cultured for 2 days and seeded with FACS into 96-well plates. UBAP2L KO single-cell clones were validated by Western blot and sequencing of PCR-amplified targeted fragment by Sanger sequencing (GATC). The sequences of gRNA are provided in the Supplementary Table 1.

All cell lines were cultured at 37 °C in 5 % CO2 humidified incubator. HeLa Kyoto and its derived cell lines were cultured in Dulbecco's Modified Eagle Medium (DMEM) (4.5 g/L glucose) supplemented with 10% fetal calf serum (FCS), 100 U/ml Penicillin + 100 µg/ml Streptomycin. U2OS cell lines were cultured in DMEM (1 g/L glucose) supplemented with 10% FCS, Non-Essential Amino Acids + Sodium Pyruvate 1 mM + Gentamicin 40 µg/mL. DLD-1 cell lines were cultured in DMEM (1 g/L glucose) supplemented with 10% FCS, 100 U/ml penicillin, and 100 µg/ml streptomycin. HCT116 cell lines were cultured in McCoy's 5 A medium + 10% FCS + 100 U/ml penicillin, and 100 µg/ml streptomycin. hTERT-RPE1 cells were cultured in Dulbecco's modified Eagle medium (DMEM) F-12 supplemented with 10% FCS, 0.01 mg/ml hygromycin B. FB 789 cell lines were cultured in MEM w/Earles's salts supplemented with 15% FCS, + Vitamines + AANE + Gentamicine 40 µg/ml + Penicillin 100 UI/ml + Streptomycin 100 µg/ml. MRC5 cell lines were cultured in DMEM (1 g/L glucose) supplemented with 10% FCS, + Gentamicine 40 µg/ml.

### Culture and differentiation of hiPSCs into neurons

Human induced pluripotent stem cells (hiPSCs, GM8330-8; kindly given by Michael E. Talkowski, Center for Genomic Medicine, Massachusetts General Hospital) were cultured in mTeSR1, following the manufacturer's instructions, and maintained on Cultrex-BME. hiPSCs were cultured at 37 °C with 5% CO2 under humidified conditions. Colonies were passaged every 3 to 4 days by manual cutting.

For the generation of inducible rtTA/Ngn2 hiPSC lines, Lentiviral vectors (PCW57) containing the human NGN2 gene cassette under the control of the TRE promoter, along with a puromycin resistance gene linked to rTetR via a T2A sequence under a constitutive promoter, were used to transduce hiPSCs. The construct was cloned and packaged into lentiviruses by the Molecular Biology Platform of the IGBMC, hereafter named TetO-NGN2 lentivirus.

hiPSCs were dissociated using Accutase and seeded in 12-well plates at a density of $6 \times 10^4$ cells per well in mTeSR1 supplemented with 10 µM Rocki (STEMCell), and incubated overnight at 37 °C. The next day, the medium was removed, and the cells were incubated for 6 h in DMEM/F12 containing 8 µg/ml Polybrene, 10 µM Rocki, and $4 \times 10^4$ TU of TetO-NGN2 lentivirus. After 6 , cells were washed with DPBS (Thermo Fisher) and returned to mTeSR1 at 37 °C and 5% CO2. Puromycin selection began on day 1 post-transduction, with 0.5 µg/ml from day 1 to 2, 1 µg/ml from day 3 to 4, and then 0.5 µg/ml again from day 5 to 7, with daily medium changes using mTeSR1. Selected hiPSC colonies were subsequently expanded in new Cultrex-BME-coated plates after Accutase dissociation and maintained under standard conditions.

For the differentiation of rtTA/Ngn2 hiPSCs into induced neurons, the differentiation was adapted from reported protocol without addition of astrocytes[81]. On day 0, TetO-NGN2 hiPSCs were dissociated with Accutase and seeded at a density of $4 \times 10^5$ cells/cm² on Poly-L-Ornithine (50 µg/cm²) + Laminin (7 µg/cm²)-coated petri dishes and in mTeSR1 supplemented with 10 µM Rocki and 3 µg/ml doxycycline. On day 1, media was replaced with DMEM/F12 + Glutamax 1X + NEAA 1X + B27 1X + N2 1X + doxycycline 3 µg/ml. On day 3, media was replaced with Neurobasal + Glutamax 1X + NEAA 1X + B27 1X + NT-3 (10 ng/mL) + BDNF (10 ng/mL) + doxycycline 3 µg/ml + cytosine β-D-arabinofuranoside (AraC) 5 µM. From day 5 to day 20, media was half-changed each 3–4 days with Neurobasal + Glutamax 1X + NEAA 1X + B27 1X + NT-3 (10 ng/mL) + BDNF (10 ng/mL) + doxycycline 3 µg/ml. Doxycycline was maintained until day 14 of differentiation. Neurons were fixed/immunostained at 20 DIV.

### Cell treatments and cell cycle synchronization

To induce the formation of the cytoplasmic AL foci by microtubule depolymerization, cells were incubated with 10 µM nocodazole (Sigma M1404-50MG) in culture media for 90 min at 37 °C.

To measure nuclear growth, cells were arrested in G1/S phase by sustained addition of thymidine (Sigma, T1895) at 2 mM. After 16 h cell were treated with 10 µM nocodazole and collected at different time points. To test the role of autophagy in AL foci abundance, cells were

incubated with 30 nM Bafilomycin (Sigma, B1793-10UG) in culture media for 6 h at 37 °C.

For cell cycle synchronizations double thymidine block and release (DTBR) protocol was used. Cells were treated with 2 mM thymidine for 16 h, washed out (three times with warm thymidine-free medium), then released in fresh thymidine-free culture medium for 8 h, treated with 2 mM thymidine for 16 h again, washed out, and then released in fresh thymidine-free culture medium for 9 h (mitosis) or 12 h early G1 phase.

## Immunofluorescence

Cells were plated on 9–15 mm glass coverslips (Menzel Glaser) in 12- or 24- well tissue culture plates. For immunofluorescence, cells were washed twice with PBS before being fixed with 4% paraformaldehyde (PFA, Electron Microscopy Sciences 15710) in PBS for 15–20 min at room temperature. Cells were washed three times for 5 min in PBS before permeabilization for 5 min with 0.5% NP-40 (Sigma) in PBS. Cells were washed three times for 5 min in PBS before blocking for 1 h at room temperature with 3% BSA in PBS-Triton 0.01% (Triton X-100, Sigma, T8787). Cells were then incubated with primary antibody in blocking buffer (3% BSA in PBS-Triton 0.01%) for 1 h at room temperature. Cells were washed three times with PBS-Triton 0.01% with gentle shaking for 10 min before being incubated with secondary antibody in blocking buffer for 1 h at room temperature in the dark. After incubation, cells were washed 3 times with PBS-Triton 0.01% for 10 min each time with gentle shaking in the dark, and then glass coverslips were mounted on glass slides using MOWIOL containing 0.75 μg/ml DAPI (Calbiochem) and imaged with 100X, 63X or 40X objectives using a Zeiss epifluorescence microscope or confocal microscope a Leica Spinning Disk Andor/Yokogawa confocal microscope.

## Live-cell imaging

For live-cell microscopy, cells were grown on 35/10 mm 4 compartment glass bottom dishes (Greiner Bio-One, 627871) or μ-Slide 8 well glass bottom (Ibidi, 80827). Before photography, cells were treated with SiR-DNA (Spirochrome, SC007) according to the manufacturer's instructions. Live cell microscopy was performed using a confocal microscope Leica/Andor/Yokogawa spinning disk, Leica CSU-W1 spinning disk, or Nikon PFS spinning disk with a 40X or 63X objectives with Live Data Mode equipped with automated temperature control.

In particular, for AL foci' dynamics assays under normal growth conditions, HeLa Kyoto GFP-Nup107 cells were analyzed by Leica CSU-W1 spinning disk (63X NA 1.4 oil objective) for 9 h. Z-stacks (10 μm range, 1 μm step) were acquired every 5 min and videos were made with maximum intensity projection images for every time point shown at a speed of 7 frames per second. For AL foci' dynamics assays after microtubule depolymerization, HeLa Kyoto GFP-Nup107 cells were arrested in S phase by thymidine block and microtubule depolymerization and nucleoporin aggregation were induced by addition of 10 μM nocodazole addition. For AL foci' dynamics assays under FXR1 knockdown (KD) or UBAP2L KD conditions, HeLa Kyoto GFP-Nup107 derived stable cell were treated with the indicated siRNAs for 48 h before imaging.

For rapid live-cell imaging, HeLa Kyoto GFP-Nup107 cells were imaged for 15 min using a Yokogawa X1 spinning disk equipped with Nikon TI2 microscope and photometrics prime 95B camera and a 100x NA 1.4 oil-immersion objective using Live SR module from GATACA system. Z-stacks spanning 3–5 μm with a step size of 1 μm were acquired every 3 s. Videos were generated from maximum-intensity projections of each time point and displayed at 7 frames per second. Quantification was performed by focusing specifically on regions adjacent to the nuclear envelope (NE). Because attachment and detachment events could not be reliably recorded across the entire nuclear circumference due to the limited field of view, each nucleus was divided into multiple NE-adjacent sections. Foci dynamics were tracked frame by frame within each section, and attachment and detachment events were recorded. Events from all sections were subsequently pooled for statistical analysis.

For AL foci and microtubules dynamics assays, cells were treated with SiR-tubulin (Spirochrome, SC002) according to the manufacturer's instructions.

For AL foci and ER' dynamics assays, HeLa Kyoto GFP-Nup107 were transfected with the pQCXIP-mScarlet-ER for 24 h and analyzed by Nikon PFS spinning disk (100X NA1.4 oil objective) for 5 h where images were acquired every 20 s (without Z-stacks) or were analyzed for 9 h where images were acquired every 2 min (without Z-stacks).

For photoactivatable nucleocytoplasmic shuttling, AID-NUP358 DLD-1 cells were transfected with the NLS-mCherry-LEXY (pDN122) for 24 h and analyzed by a Leica Sp5 confocal inverted microscope (DMI6000) system (40X NA1.3 oil objective). The circular region of interest (ROI; ~38 μm²) was placed onto single, selected cells. The ROI was scanned with a 458-nm laser beam for 10 min followed by a 20-min dark-recovery phase. The mCherry signal was imaged in parallel every 10 s for 30 min using the 561-nm laser line for excitation.

For photoconversion assays, HeLa Kyoto mEos2-Nup133 cells were analyzed by Yokogawa X1 spinning disk equiped with Nikon TI2 microscope, photometrics prime 95B camera and ILas Head for photomanipulation (100X NA1.4 oil objective). The modular software part developed by the GATACA system was used to set the acquisition of the selected area to be photoconverted with applied repeats of blue light for a period of 200 s (typical value, depending on the size of the area) by the 500 mW laser diode at 100%. A 30-s acquisition pre-sequence and with acquisition every 10 s in the green and red channels has been performed. Two post sequences were carried out after acquisition, the first for 30 s with an interval every 10 s in order to quickly observe the effect of photoconversion and the second sequence for 12 h every 15 min to see the effect over a longer period. The software was set up to use spinning with the 491 nm 50 mW laser diode at 20% power with an acquisition time of 150 ms for the green channel and the 561 nm 50 mW laser diode at 30% power with 150 ms exposure time for the red channel.

## Single molecule localization microscopy

For super-resolution imaging using SMLM, cells were plated on 35 mm glass bottom dishes with a 14 mm micro-well #1.5 cover glass (Cellvis). Samples were prepared as described above for immunofluorescence but were not mounted using MOWIOL and were stored in PBS.

The imaging was performed as previously described, using the splitSMLM method[82] for multi-color SMLM based on a dichroic image splitter. In brief, the immunolabeled samples were mounted in an imaging buffer containing 200 U/ml glucose oxidase (G2133, Sigma), 1000 U/ml catalase (C1345, Sigma), 10% w/v glucose, 200 mM Tris-HCl pH 8.0, 10 mM NaCl, 50 mM MEA (30080, Sigma), with addition of 2 mM COT (138924, Sigma). The buffer optimizes the performance of the fluorophores AF647, CF660C and CF680 for three-color direct stochastic optical reconstruction microscopy (dSTORM). Imaging was then performed on a modified Leica SR GSD system, with a HCX PL APO 160X/1.43 Oil CORR TIRF PIFOC objective. The acquisitions began with a pumping phase, during which the sample was illuminated with the 642 nm 500 mW fiber laser (MBP Communication Inc.) but the fluorescence was not recorded due to a very high density of fluorophores in a bright state. The image collection started when the density of fluorophores dropped to a level that allowed observation of individual molecules and in case of emitter depletion, the sample was additionally illuminated with a 405 nm diode laser (Coherent Inc.) to increase the emitter density ("back-pumping"). The resulting emission was split with an Optosplit II (Cairn Research) image splitter, attached

to a camera port of the microscope, and containing a Chroma T690LPXXR dichroic mirror. This creates a long and short wavelength channel on the Andor iXon Ultra 897 EMCCD camera attached to the image splitter. Each fluorophore thus shows a characteristic brightness ratio between the two channels, enabling the separation of spectrally different fluorophore species (i.e., spectral demixing)[82]. Localization of the single molecules in both channels was performed using the Leica LAS X software with the "direct fit" fitting method. The localization tables were then exported and further processed using the SharpViSu software[29,31,83] and the SplitViSu plugin[82]. This accounts for spectral demixing, correction of chromatic errors, drift-correction, and the reconstruction of super-resolution images. A pixel size of 10 nm was used to reconstruct the super-resolution images from 2D histograms of single-molecule coordinates. For the single-color imaging without an image splitter, the images were recorded with an Andor iXon + EMCCD camera, attached to the second camera port of the microscope. Data were then processed as described, using the Leica LAS X and SharpViSu software, while omitting the spectral demixing steps in the SplitViSu plugin.

The magnified NPC regions shown in the article were upsampled by bilinear interpolation for improved visualization.

### Correlative light and electron microscopy

Cells were grown on carbon coated sapphire disks (3 mm diameter, 50 μm thickness, Wohlwend GmbH, art. 405) and high pressure frozen (HPM010, AbraFluid) in their culture medium. Freeze substitution was performed in a Leica EM-AFS2 with 0.1 % uranyl acetate in dry acetone for 24 h at −90 °C. The temperature was then raised to −45 °C over 9 h (slope 5 degrees / hour) and the samples were further incubated at this temperature for 5 h to increase electron contrast. After rinsing in acetone, the samples were infiltrated with increasing concentrations of Lowicryl HM20 (Polysciences Inc.), while raising the temperature to −25 °C. Finally, the blocks were polymerized using UV light. After removal of the sapphire disks from the block surface, 300 nm sections were cut parallel to the block surface using an ultramicrotome (UC7, Leica Microsystems). The sections were collected on carbon coated mesh grids (S160, Plano).

Fluorescence imaging of the sections was carried out with a widefield microscope (Olympus IX81), equipped with a 100×1.40 NA Plan-Apochromat objective, placing the grids in water in a glass-bottom dish (Mattek). After light microscopy acquisition, the grids were post-stained with 2% uranyl acetate in 70% methanol and Reynold's lead citrate. Electron microscopy (EM) was performed using a Tecnai F30 transmission electron microscope (Thermo Fisher) at 300 kV acceleration voltage. The grid squares that were previously imaged at the light microscope (LM) were mapped at low mag. After registration with the LM image, TEM tomography was performed on the areas of interest using the software package SerialEM[84]. Tomograms were reconstructed with IMOD[85]. Correlation between LM and EM images was done with the plugin ec-CLEM[86] of the software platform Icy[87], using features of the samples that could be identified in both imaging modalities.

### Image analysis

Image quantification analysis was performed using ImageJ or CellProfiler software. In particular, the presence of Nups AL foci was visually confirmed by two individuals.

For nuclear size or nuclear envelope intensity analysis of Nups, CellProfiler software was used. Briefly, the Prewitt Edge Finder method was used to enhance the edges in a DAPI image, and then a nucleus size threshold was set to automatically identify nuclei. This allowed identification and measurement of nuclei area, morphology factor and mean nuclear intensity for the desired channel. The software's parameter measurements were exported to an Excel file and statistically analyzed.

For the size of Nups foci, ImageJ software was used. In brief, the Nups channel was thresholded in ImageJ and then the size of the Nups foci was calculated by analyzing particles (size 1-100). AL foci that were considered "large" had a size bigger than $0.3\,\mu m^2$.

For co-localization analysis of RanBP2 and Nups in the cytoplasm, CellProfiler software was used. The Prewitt Edge Finder method was used to enhance the edges in a DAPI image, and then nuclei, RanBP2 and different Nups were identified according to the corresponding channels. The cytoplasmic fraction of RanBP2 and Nups was identified using the MaskObjects module. Co-localization analysis of RanBP2 and Nups in cytoplasm was conducted by the MeasureObjectOverlap module. The software's parameter measurements were exported to an Excel file and statistically analyzed. For nuclear envelope or cytoplasm intensity of Ran, ImageJ was used. The ROIs of the cell and nuclei were selected based on tubulin and DAPI channels, and the ROI of the cytoplasm was selected by the XOR function of cell and nuclei. Finally, the mean gray value of the nuclei and cytoplasm was measured.

For AL-NPCs analysis in the cytoplasm, AL-NPCs were recognized by the Nup133 signal surrounding Nup62 signals. The number of AL-NPCs was counted visually by two individuals.

For the measurement of NPCs density on the nuclear membrane, ImageJ software was used. Briefly, the images of the Nup62 channel were converted to 32 bits using ImageJ and then filtered using Gaussian blur (sigma = 2). The nuclear region was manually selected and then the 'Find Maxima' function (noise tolerance 10) was performed to obtain the area of the selected region and the number of AL-NPCs.

For protein import and export assay analysis, ImageJ software was used. Briefly, the ROI within the uniform distribution of NLS-mCherry-LEXY signal was selected in the nuclear region, and the measurement of the mean gray value of the same ROI in each frame (sequential time points) was performed by the multi-measure option.

For calculation of the mean fluorescence intensity of Nups as a function of distance from the nucleus a custom Python script was used with help of the scikit-image module[88]. The nucleus and the nuclear envelope were identified using the RanBP2 channel following well-established procedures[89,90]. Similarly, the region corresponding to the ER was determined from the mScarlet-ER channel. Regions corresponding to RanBP2 foci were detected using the scikit-image method blob_log for blob detection. The analyzed region for Climp63 and mScarlet-ER corresponded to the ER region excluding the nucleus. The analyzed region for the RanBP2 foci corresponded to the blob regions that fall within the ER region excluding the nucleus. For each pixel in the analyzed regions, the minimum Euclidean distance to any nuclear envelope pixel was taken as the distance from the nucleus. Given the intensity-distance pairs for each pixel, the pixels were binned for the distance and the mean pixel intensity determined for each bin. The obtained histograms were finally normalized for the purpose of comparison.

For colocalization analysis, to assess pixel colocalization/correlation, we used correlation measurement within CellProfiler. Briefly, we used the "MaskImage" module to obtain Climp63, mScarlet-ER, and RanBP2 located in the cytoplasm. "Measure Colocalization" modules were used to study the colocalization and correlation between intensities in different images (different color channels) on a pixel-by-pixel basis across an entire image. The number of cells measured per condition was listed in the corresponding figure's legend.

For nuclear height measurement, confocal microscopy was used to perform z-sampling of the DAPI channel of the sample, with a z-step of 0.5 μm. Finally, Fiji analysis was used to obtain an approximate nuclear height based on the Z-Extent.

### Protein sample preparation and western blotting

After centrifugation at 250 g for 3 min at 4 °C, cells were washed twice with cold PBS, and lysates were prepared using cell lysis buffer (50 mM Tris-HCl pH 7.5, 150 mM NaCl, 0.5% sodium deoxycholate, 0.1% SDS,

0.5% sodium deoxycholate, 1% Triton X-100, 1 mM EDTA, 1 mM EGTA, 2 mM Sodium pyrophosphate, 1 mM Na$_3$VO4 and 1 mM NaF) supplemented with protease inhibitor cocktail (Roche) and incubated on ice for 30-60 min. After centrifugation at 16,000 $g$ for 15 min at 4 °C, the supernatant was transferred to the new clean Eppendorf tubes, and the total protein concentration was measured with a Bio-Rad Protein Assay kit (Bio-Rad). Separation of cytoplasmic and nuclear fractions was done using the NE-PER nuclear and cytoplasmic extraction reagent kit (Thermo Scientific™, 78833). Protein samples were boiled for 10 min at 95 °C in 1X Laemmli buffer (LB) with β-Mercaptoethanol (BioRad, 1610747).

SDS-PAGE of proteins was done using pre-cast 4–12% Bis-Tris gradient gels (Thermo Scientific, NW04120BOX) or pre-cast NuPAGE™ 3–8% Tris-Acetate gradient Gels (Thermo Scientific, EA0378BOX). Proteins were transferred to a polyvinylidene difluoride (PVDF) membrane (Millipore, IPFL00010) using wet transfer modules (BIO-RAD Mini-PROTEAN® Tetra System). Membranes were blocked in 5% bovine serum albumin (BSA, Millipore, 160069), or 5% non-fat milk powder mixed with 3% BSA resuspended in TBS-T for 1 h at room temperature, followed by incubation with primary antibodies diluted in TBS-T 5% BSA/5% milk overnight at 4 °C or 1 h at room temperature and secondary antibodies diluted in TBS-T 5% BSA/5% milk at room temperature. Membranes were imaged using SuperSignal West Pico (Pierce, Ref. 34580). Western blotting images were acquired by GE Healthcare Amersham Imager 600 or Invitrogen iBright 1500. The grayscale value of protein bands was quantified using ImageJ software.

## Immunoprecipitation

Cell lysates were prepared for immunoprecipitations (IP) as described above. Lysates were adjusted to equal volume and concentration. For endogenous IP experiments, lysates were incubated with IgG or target-specific antibodies overnight at 4 °C with rotation, and then incubated with protein G sepharose 4 Fast Flow beads (GE Healthcare Life Sciences) for 4 h at 4 °C with rotation. Before using, beads were blocked with 3% BSA diluted in 1X cell lysis buffer and incubated for 2 h at 4 °C with rotation. The incubated IgG/specific antibodies-samples-beads were washed with washing buffer (25 mM Tris-HCl pH 7.5, 300 mM NaCl, 0.5% Triton X-100, 0.5 mM EDTA, 0.5 mM EGTA, 1 mM Sodium pyrophosphate, 0.5 mM Na$_3$VO$_4$ and 0.5 mM NaF) or TBS-T 4 to 6 times.

For GFP-IP/HA-IP experiments, GFP-Trap A agarose beads (Chromotek) or Pierce™ anti-HA Magnetic Beads (Thermo Scientific 88836) were used. Cells expressing GFP- or HA-tagged plasmids for at least 24 h before lysis were used. Lysates were incubated with beads for 4 h at 4 °C with rotation, and then washed with washing buffer (25 mM Tris-HCl pH 7.5, 300 mM NaCl, 0.5% Triton X-100, 0.5 mM EDTA, 0.5 mM EGTA, 1 mM Sodium pyrophosphate, 0.5 mM Na$_3$VO$_4$ and 0.5 mM NaF) or TBS-T 4 to 6 times.

## Gel fractionation

To calibrate the Superose 6 Increase 3.2/300 column, the gel filtration standard (Bio-Rad, Catalog # 1511901) was rehydrated and diluted to 1.5 mg/mL according to the instruction of the manufacturer. The column was equilibrated using buffer comprising 15 mM pH 7.5 Tris, 15 mM NaCl, 60 mM KCl, 340 mM sucrose, 0.15 mM spermine, 0.5 mM spermidine, 0.5% Triton X-100, 1 mM NaF, 10 mM DTT. The standard sample (50 μL) was separated by a flow rate of 0.015 ml/min. Kav values for the standards were calculated. The calibration curve of Kav versus the logarithm of their molecular weights was determined as follows:

$$Kav = \frac{Ve - Vo}{Vt - Vo}$$

where Ve = elution volume of the proteinVo = column void volume = 0.8 mLVt = total bed volume = 2.4 mL

HeLa cell cytosolic and nuclear fractions were prepared as described above. For the gel filtration of the HeLa cell cytosol extraction sample, a 2.4-mL Superose 6 Increase 3.2/300 column (Cytiva) was equilibrated using buffer comprising 15 mM pH 7.5 Tris, 15 mM NaCl, 60 mM KCl, 340 mM sucrose, 0.15 mM spermine, 0.5 mM spermidine, 0.5% Triton X-100, 1 mM NaF, 10 mM DTT. For the gel filtration of the HeLa cell nuclear fraction sample, a 2.4-ml Superose 6 Increase 3.2/300 column (Cytiva) was equilibrated using buffer comprising 50 mM Tris 7.5, 1% NP-40, 150 mM NaCl, 1 mM EDTA, 0.1 % SDS, 0.5% sodium deoxycholate. The cytosolic and nuclear fractions (typically 1–3 mg/ml) were clarified by centrifugation for 15 min at 12,000 × $g$ before loading to the equilibrated gel filtration column. Samples were separated by a flow rate of 0.017 ml/min, and 50 μL fractions were collected for Western blot analysis (typically collects from 1.2 mL to 2.4 mL).

## LC-MS/MS analysis

MS grade Acetonitrile (ACN), MS grade H2O and MS grade formic acid (FA) were from ThermoFisher Scientific (Waltham, MA, USA). Sequencing-grade trypsin/Lys C mix was from Promega (Madison, WI, USA). Trifluoroacetic acid (TFA) and ammonium bicarbonate (NH4HCO3) were from Sigma-Aldrich (Saint-Louis, MO, USA).

For sample preparation prior to LC-MS/MS analysis beads from immunoprecipitation experiments were incubated overnight at 37 °C with 20 μL of 50 mM NH4HCO3 buffer containing 1 μg of sequencing-grade trypsin/Lys C mix. The digested peptides were loaded and desalted on evotips provided by Evosep (Odense, Denmark) according to manufacturer's procedure before LC-MS/MS analysis. For LC-MS/MS acquisition samples were analyzed on a timsTOF Pro 2 mass spectrometer (Bruker Daltonics, Bremen, Germany) coupled to an Evosep one system (Evosep, Odense, Denmark) operating with the 30SPD method developed by the manufacturer. Briefly, the method is based on a 44-min gradient and a total cycle time of 48 min with a C18 analytical column (0.15 × 150 mm, 1.9 μm beads, ref EV-1106) equilibrated at 40 °C and operated at a flow rate of 500 nL/min. H2O/0.1% FA was used as solvent A and ACN/ 0.1% FA as solvent B. The timsTOF Pro 2 was operated with a DIA-PASEF method comprising 12 pydiAID frames with 3 mass windows per frame resulting in a cycle time of 0.975 s as described in Bruker application note LCMS 218.

For data analysis MS raw files were processed using Spectronaut 18 (Biognosys, Switzerland). Data were searched against the SwissProt Homo Sapiens database (downloaded 07 2023, 20423 entries). Specific tryptic cleavages were selected and a maximum of 2 missed cleavages were allowed. The following post-translational modifications were considered for identification: Acetyl (Protein N-term) and Oxidation (M) as variables. Identifications were filtered based on a 1% precursor and protein Qvalue cutoff threshold. The protein LFQ method was set to automatic and the quantity was set at the MS2 level with a cross-run normalization applied. Multivariate statistics on protein measurements were performed using Qlucore Omics Explorer 3.9 (Qlucore AB, Lund, SWEDEN). A positive threshold value of 1 was set to allow a log2 transformation of abundance data for normalization i.e., all abundance data values below the threshold are replaced by 1 before transformation. The transformed data were finally used for statistical analysis i.e., the evaluation of differentially present proteins between two groups using a bilateral Student's $t$ test. A p-value better than 0.05 was used to filter differential candidates.

## Data acquisition, statistical analysis and reproducibility

All experiments were performed in a strictly double-blinded manner. All experiments were performed with at least three independent biological replicates, unless stated otherwise in the figure legends and image quantifications were carried out in a blinded manner. Curves and graphs were made using GraphPad Prism and Adobe Illustrator software. Data were analyzed using a one-sample two-tailed T-test or a two sample unpaired two-tailed T-test (two-group comparison or folds

increase relative to the control, respectively) or one-way ANOVA. A p-value less than 0.05 (typically ≤ 0.05) was considered statistically significant and stars were assigned as follows: *$P < 0.05$, **$P < 0.01$, ***$P < 0.001$, ****$P < 0.0001$.

## Reporting summary

Further information on research design is available in the Nature Portfolio Reporting Summary linked to this article.

## Data availability

Source data supporting the findings of this study are provided with this paper. All data necessary to interpret, verify, and extend the results reported in this article, including source data underlying all quantitative analyses, proteomic data and graphs, are included in the Source Data files. The mass spectrometry proteomics data have been deposited to the ProteomeXchange Consortium via the PRIDE partner repository with the dataset identifier PXD073946. Representative microscopy images and videos (spinning-disk confocal live imaging, super-resolution microscopy, and EM/CLEM) are included in the figures and Supplementary Information. Due to the large size and complexity of the raw imaging datasets (>50 GB), full-resolution microscopy data are not deposited in a public repository. Raw imaging data are available from the corresponding author upon request. Source data are provided with this paper.

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

## Acknowledgements

We thank members of the Sumara, I. and Ricci, R. laboratories for helpful discussions on the manuscript and IGBMC facilities for their help. We are grateful to Valérie Doye, Bernhard Hampoelz and participants of the "Rembo" workshop for stimulating discussions on data interpretation. We are grateful to Arnaud Echard and Anakine Prizins for helpful support on photoconversion experiments and comments on the manuscript. We are grateful to Mary Dasso, Julie Eichler, Juliette Godin, André Hoelz, Ralph Kehlenbach, Luca Proietti, Romeo Ricci, Thierry Seroz, Michael E. Talkowski, Catherine-Laure Tomasetto and Zhirong Zhang for generously sharing reagents. We acknowledge the Virus and Molecular Biology platform of the IGBMC for the lentivirus production. We acknowledge the IGBMC imaging center, member of the national infrastructure France-BioImaging supported by the French National Research Agency (ANR-10-INBS-04). Lin, J. Liu, X. and Liao, Y. were supported by PhD fellowships from the China Scholarship Council (CSC). Agote-Aran, A., was supported by PhD fellowship from the IMC-Bio graduate school and a fellowship from the "Ligue Nationale Contre le Cancer". Cloarec, M. was supported by PhD fellowship from the doctoral school University of Strasbourg. Andronov, L., Schoch, R. and Klaholz, B. P. acknowledge support by Agence National de la Recherche (ANR), Institut National du Cancer (INCa), the Integrated Structural Biology platform (IGBMC-CBI) and by the French Infrastructure for Integrated Structural Biology (FRISBI) ANR-10-INSB-05-01, Instruct-ERIC and iNEXT-Discovery. Golzio, C. is a permanent INSERM investigator. Lemée, M.V. was a doctoral fellow supported by IMC-Bio funds and Fondation de France (WB- 2022-45868). This work was funded by ANR under the projects (ANR-22-CE12-0011, to Golzio, C. and ANR-22-CE13-0025, NICE4Nups to Sumara, I. and Klaholz, B.P.) Research in Sumara, I. laboratory was supported by the grant ANR-10-LABX-0030-INRT, a French State fund managed by ANR under the frame program Investissements d'Avenir ANR-10-IDEX-0002-02, IGBMC, CNRS, ARC, INCa (PLBIO 2022-082), and AFM-Telethon.

## Author contributions

The role of RanBP2 in human cytosolic NPCs was initially discovered by A.A.A. and confirmed by J.L., and the research plan and the conceptualization of the results were based on discussions between J.L. and I.S. J.L. performed most of the experiments and analyzed the data. Y.L. performed several experiments on UBAP2L and M.C analyzed the role of additional ER factors. L.A. and R.S. helped with splitSMLM analyses, P.R. helped on and supervised CLEM analysis, R.Z. helped with gel-filtration chromatography under supervision of M.R. E.G. helped with imaging, photoconversion experiments and image analysis. Mass spectrometry analysis was performed by V.C. under supervision of G.C. M.V.L. generated the hiPSC derived neurons under supervision of C.G. X.L and M.C. analyzed FB789 and MRC5 cell lines and C.K. helped with the reagents and materials. G.C., Y.S., and B.P.K. helped with the design of the experiments and supervision. I.S. supervised the project and J.L. and I.S. wrote the manuscript with input from all authors.

## Competing interests

The authors declare no competing interests.
