## [Transparent Peer Review file · Nature Communications]

RanBP2-dependent annulate lamellae drive nuclear pore assembly and nuclear expansion

Corresponding Author: Dr Izabela Sumara

This manuscript has been previously reviewed at another journal. This document only contains information relating to versions considered at Nature Communications. Mentions of the other journal have been redacted.

Version 0:

Reviewer comments:

Reviewer #1

(Remarks to the Author)

This is a revised manuscript, transferred to Nature Communications after review and consideration at [REDACTED]. The original manuscript was extensive, robust and used state of the art microscopical approaches and orthogonal methods of intervention and observation to reveal the role for RanBP2 in clustering and incorporating cytoplasmic aggregates of NPCs in Annulate Lamellae (AL) into the primary nucleus. In revision, the authors have addressed all the Nat Cell Biol comments and incorporated substantial amounts of new data into the manuscript - whilst it was a mammoth story before, it is doubly-mammoth here! This is a massive amount of work and the authors ought be lauded for their efforts here. Whilst the narrative and structure has improved substantially since the original submission, it is quite a challenging manuscript to carry in your head whilst reading. I think the manuscript is suitable for publication.

Regards my specific comments, the authors have demonstrated physiological contribution of the AL-insertion pathway to populating the interphase nucleus with NPCs and allowing the nucleus to expand. The new timelapse imaging of AL-insertion into the NE is a strong addition, as is the separation of Nup153/ELYS effects from a RanBP2-dependent insertion. They have clarified issues about the RanBP2-CLIMP63 interaction, and whilst not showing it is direct, they have included controls to highlight the specificity of the co-precipitation and made it clearer how the IP was performed.

Mechanistically, the mapping of the AL-clustering activity to the FG domain of RanBP2 is a helpful addition.

The G1-restricted RanBP2 contribution to nuclear expansion is interesting, but I wonder if the authors could explain how the cells remain in G1 for the imaging period (48h); from the methods, the thymidine arrest is only for 16h, then dmsoc/noc for 90 minutes and then imaging. The restriction on nuclear expansion is clear, but I can't work out why the cells don't reenter the cycle and divide over the next 2 days? The details of this assay could be added to the methods as I don't think this is included.

A second element that could be clarified in the text a little is the role of RanBP2 in oligomerising Y-complexes within individual NPCs and binding Nup93 (Fig 6E-G) from the RanBP2-dependent clustering of individual NPCs into larger AL aggregates. I think these are two separate things, and I'm not sure that the Y-complex and Nup93 binding is needed for the narrative.

Lastly, the cartoon at the end really helped with understanding; however, it is a little stylised. Rather than a grey block for the ER, I wonder if individual INM, ONM and ER membranes could be depicted. It is a little hard to understand what the authors think is happening to accommodate this insertion. There will certainly be some gymnastics required to place these AL-NPCs across the INM/ONM for contribution to transport activity, but it isn't clear from the cartoon how this would happen. To help others follow up on this work, I would encourage the authors to outline how this might happen so that subsequent experimental followups can be planned.

Minor

L214, underlying nuclear membrane should probably be underlying nuclear lamina.

Reviewer #2

(Remarks to the Author)

Overall assessment of revision: The revised manuscript addresses my major concerns with the initial submission. The most critical aspect related to the membrane topology of the nuclear envelope at sites that the authors interpret as insertion events for AL into the NE. The new data acquired using splitSMLM that suggests openings in the nuclear envelope at the sites of AL integration is the most critical. While this addresses my major concern it should be more robustly presented (see below). In addition, the cartoon model should incorporate these new insights. One other issue related to the newly added data relates to interpretation of the described cell cycle differences in Ran distribution.

Specific Points:

1. Related to the gaps in the nuclear envelope: In Fig. 3c it would be helpful to demarcate the edges of the PDI signal that the authors interpret as gaps. Because of the importance of this finding, either a quantitative representation of how commonly this was observed and/or additional examples should be included – even if just from the timepoint where the gap and apposed AL are viewable.
2. The cartoon model in 7k still fails to capture the topological requirements for the incorporation of AL into the NE – this needs to be addressed as for non-experts this will be how they conceive of the study's conclusions. As drawn the embedding of the NPC into the two lipid bilayers needs to be accurately represented and the NE must have a gap into which the NPC could move from the AL. The intermediate steps should be shown.
3. The authors have added new data assessing the distribution of the Ran GTPase and interpret the findings as functionally relevant. However, this line of investigation is too speculative at this stage. The authors discuss a "gradient" of Ran, but they are looking only at Ran distribution. The Ran gradient relates to the nucleotide state of Ran-GTP and is a sum effect of Ran, the RanGEF RCC1 and RanGAP. If this is to be interpreted it is essential to understand if the Ran GTPase gradient (of which there are biosensors) is actually cell cycle regulated. Also, it would be important to assess if Ntf2 (the dedicated transport factor for Ran) levels are different – but Ntf2 is not even mentioned in the manuscript. Last, Ntf2 is particularly relevant to the study because it has previously been tied to nuclear size scaling, which the authors report to be affected (PMIDs: 20946986, 26823604).

Reviewer #3

(Remarks to the Author)

In this revised submission, the authors have improved the manuscript, where particularly the splitSMLM data in Fig. 3 are a convincing complement to demonstrate fusion of NPCs from cytoplasm to NE. Overall, with reference to my previous report on this manuscript, I'd have a few remaining comments/questions, the responses to which, as far as I am concerned may next be assessed by the editor.

2nd paragraph of Results section: CLEM analysis "showed these foci to be localized ..., demonstrating that they are AL (Fig. 1a)". The amount of images shown is too small to be conclusive about this. More data would need to be shown and ideally quantified to support this conclusion. Alternatively, this statement should be toned down to a less conclusive, more suggestive variant.

Related to this, the caption of Fig. 1a denotes the shown CLEM images as being "representative". At the risk of further increasing the length of this already voluminous submission: Could the authors complement these images with additional biological repeats in an SI Figure, to demonstrate that these observations can at least be made in triplicates? That way, even when full quantification may be too ambitious, the authors would not need to state anything about representativeness of the shown image, but can just demonstrate it with data.

A similar observation can be made about the CLEM data in Fig. 6a.

In the section "AL are highly dynamic and can fuse with the NE...", its title, main text and figure labelling: Fig. 2e and SI videos 2,3 are said to show that there are fusion and fission events of AL foci with the NE. The shown images, even for Z slices in confocal microscopy, at this scale still represent 2D projections of a 3D reality, which may complicate interpretation. Is it obvious, depending on the depth resolution in these experiments, that the foci are not drifting behind/in front of the view of the nucleus?

What data is the quantification in Fig. 2d based on? Having looked at Fig. 2e and SI videos 2,3, I struggle to identify the numbers of attachment/detachment events noted in Fig. 2d, and from the underpinning data shown, it seems that their identification is not that unambiguous. Could the authors explain more explicitly how this (per se very welcome) quantification results from the microscopy data?

Data in Fig. 2f are shown to further substantiate the claim of AL-insertion into NE, albeit with the authors' caveat about long-term stability and brightness of the fluorophores in these data. Is there a negative control for these experiments?

Minor:

Fig. 1b, top-left panel: A "2" label appears to be missing in this image.

Point-by-point response to Reviewers' Comments

Reviewer #1

This is a revised manuscript, transferred to Nature Communications after review and consideration at [REDACTED]. The original manuscript was extensive, robust and used state of the art microscopical approaches and orthogonal methods of intervention and observation to reveal the role for RanBP2 in clustering and incorporating cytoplasmic aggregates of NPCs in Annulate Lamellae (AL) into the primary nucleus. In revision, the authors have addressed all the Nat Cell Biol comments and incorporated substantial amounts of new data into the manuscript - whilst it was a mammoth story before, it is doubly-mammoth here! This is a massive amount of work and the authors ought be lauded for their efforts here. Whilst the narrative and structure has improved substantially since the original submission, it is quite a challenging manuscript to carry in your head whilst reading. I think the manuscript is suitable for publication.

We thank Reviewer 1 for their very positive assessment of the revised manuscript and their recognition of the extensive work involved. In response to the comments, we have made further efforts to improve clarity by simplifying the narrative and structure and by distributing the data across separate figures (now nine main figures). We appreciate the reviewer's support for the publication of this study.

Regards my specific comments, the authors have demonstrated physiological contribution of the AL-insertion pathway to populating the interphase nucleus with NPCs and allowing the nucleus to expand. The new timelapse imaging of AL-insertion into the NE is a strong addition, as is the separation of Nup153/ELYS effects from a RanBP2-dependent insertion. They have clarified issues about the RanBP2-CLIMP63 interaction, and whilst not showing it is direct, they have included controls to highlight the specificity of the co-precipitation and made it clearer how the IP was performed.

Mechanistically, the mapping of the AL-clustering activity to the FG domain of RanBP2 is a helpful addition.

We thank Reviewer 1 for these comments and for acknowledging the additional mechanistic insights provided in the revised manuscript.

The G1-restricted RanBP2 contribution to nuclear expansion is interesting, but I wonder if the authors could explain how the cells remain in G1 for the imaging period (48h); from the methods, the thymidine arrest is only for 16h, then dmsu/noc for 90 minutes and then imaging.

The restriction on nuclear expansion is clear, but I can't work out why the cells don't reenter the cycle and divide over the next 2 days? The details of this assay could be added to the methods as I don't think this is included.

We thank the reviewer for pointing this out and apologize for the lack of clarity. In these experiments, cells were maintained at the G1/S boundary by sustained thymidine treatment, followed by microtubule depolymerization or RanBP2 depletion, which both prevent nuclear growth (Fig. 3g-i and Supplementary Fig. 9). We have updated the experimental schemes, the Results text, and the corresponding figure legends to clarify this point.

A second element that could be clarified in the text a little is the role of RanBP2 in oligomerising Y-complexes within individual NPCs and binding Nup93 (Fig 6E-G) from the RanBP2-dependent clustering of individual NPCs into larger AL aggregates. I think these are two separate things, and I'm not sure that the Y-complex and Nup93 binding is needed for the narrative.

We agree with Reviewer 1. We have revised the text to clearly distinguish these two processes and to emphasize that RanBP2 plays distinct roles during AL assembly. Under physiological conditions, RanBP2 contributes to scaffold assembly within individual AL-NPCs, including organization of Y-complex components and interactions with Nup93, whereas under stress conditions it promotes the expansion and clustering of AL-NPCs in the cytoplasm.

Lastly, the cartoon at the end really helped with understanding; however, it is a little stylised. Rather than a grey block for the ER, I wonder if individual INM, ONM and ER membranes could be depicted. It is a little hard to understand what the authors think is happening to accommodate this insertion. There will certainly be some gymnastics required to place these AL-NPCs across the INM/ONM for contribution to transport activity, but it isn't clear from the cartoon how this would happen. To help others follow up on this work, I would encourage the authors to outline how this might happen so that subsequent experimental followups can be planned.

We thank the reviewer for these helpful suggestions and agree that the previous schematic model did not sufficiently reflect the complexity of the underlying data. We have therefore prepared a revised model that explicitly depicts the inner nuclear membrane (INM), outer nuclear membrane (ONM), and endoplasmic reticulum (ER), and illustrates how AL-NPC insertion across these membranes may occur. In addition, the revised schematic model highlights open questions to guide future experimental investigations. This updated model is now presented in Fig. 9.

Minor

L214, underlying nuclear membrane should probably be underlying nuclear lamina.

We thank the reviewer for pointing out this error, which has now been corrected.

Reviewer #2

Overall assessment of revision: The revised manuscript addresses my major concerns with the initial submission. The most critical aspect related to the membrane topology of the nuclear envelope at sites that the authors interpret as insertion events for AL into the NE. The new data acquired using splitSMLM that suggests openings in the nuclear envelope at the sites of AL integration is the most critical. While this addresses my major concern it should be more robustly presented (see below). In addition, the cartoon model should incorporate these new insights. One other issue related to the newly added data relates to interpretation of the described cell cycle differences in Ran distribution.

We thank Reviewer 2 for their assessment of the revised manuscript and for highlighting the importance of the new splitSMLM data. In response, we have strengthened the presentation of these data and revised the schematic model to incorporate the new insights. The updated model is now shown in Fig. 9. We also address the comments regarding the interpretation of cell cycle-dependent Ran distribution below.

Specific Points:

1. Related to the gaps in the nuclear envelope: In Fig. 3c it would be helpful to demarcate the edges of the PDI signal that the authors interpret as gaps. Because of the importance of this finding, either a quantitative representation of how commonly this was observed and/or additional examples should be included – even if just from the timepoint where the gap and apposed AL are viewable.

We thank the reviewer for this suggestion. We have included additional examples in Fig. 3c showing gaps in the PDI signal with apposed annulate lamellae to further support this observation.

2. The cartoon model in 7k still fails to capture the topological requirements for the incorporation of AL into the NE – this needs to be addressed as for non-experts this will be how they conceive of the study's conclusions. As drawn the embedding of the NPC into the

two lipid bilayers needs to be accurately represented and the NE must have a gap into which the NPC could move from the AL. The intermediate steps should be shown.

We thank the reviewer for this comment. As outlined above, we have prepared a revised schematic model that explicitly incorporates the required nuclear envelope topology, including membrane gaps, accurate representation of the inner and outer nuclear membranes, and intermediate steps of AL-NPC incorporation into the nuclear envelope. This updated model is now presented in Fig. 9.

3. The authors have added new data assessing the distribution of the Ran GTPase and interpret the findings as functionally relevant. However, this line of investigation is too speculative at this stage. The authors discuss a “gradient” of Ran, but they are looking only at Ran distribution. The Ran gradient relates to the nucleotide state of Ran-GTP and is a sum effect of Ran, the RanGEF RCC1 and RanGAP. If this is to be interpreted it is essential to understand if the Ran GTPase gradient (of which there are biosensors) is actually cell cycle regulated. Also, it would be important to assess if Ntf2 (the dedicated transport factor for Ran) levels are different – but Ntf2 is not even mentioned in the manuscript. Last, Ntf2 is particularly relevant to the study because it has previously been tied to nuclear size scaling, which the authors report to be affected (PMIDs: 20946986, 26823604).

We agree with Reviewer 2 and have toned down the interpretation of these data accordingly. We have revised the text to explicitly state that our analysis addresses Ran localization rather than the Ran-GTP gradient. We now clarify that the canonical Ran-GTP gradient depends on the coordinated activities of RCC1, RanGAP, and the nucleotide state of Ran, and not solely on Ran distribution. In addition, we now discuss the role of Ran-associated factors, noting that the Ran-specific transport factor NTF2, which has been implicated in nuclear size scaling was not directly examined in this study. We emphasize that whether cell cycle-dependent regulation of these components contributes to the observed changes in Ran distribution remains an open question.

Reviewer #3

In this revised submission, the authors have improved the manuscript, where particularly the splitSMLM data in Fig. 3 are a convincing complement to demonstrate fusion of NPCs from cytoplasm to NE. Overall, with reference to my previous report on this manuscript, I'd have a few remaining comments/questions, the responses to which, as far as I am concerned may next be assessed by the editor.

We thank Reviewer 3 for their positive assessment of the revised manuscript and for recognizing the value of the splitSMLM data in Fig. 3. We are happy to address the remaining comments and questions below.

2nd paragraph of Results section: CLEM analysis "showed these foci to be localized ..., demonstrating that they are AL (Fig. 1a)". The amount of images shown is too small to be conclusive about this. More data would need to be shown and ideally quantified to support this conclusion. Alternatively, this statement should be toned down to a less conclusive, more suggestive variant.

We thank the reviewer for this comment. We have toned down the statement to a more suggestive wording to avoid overinterpretation and have added additional representative CLEM examples to further support the identification of these structures as annulate lamellae.

Related to this, the caption of Fig. 1a denotes the shown CLEM images as being "representative". At the risk of further increasing the length of this already voluminous submission: Could the authors complement these images with additional biological repeats in an SI Figure, to demonstrate that these observations can at least be made in triplicates? That way, even when full quantification may be too ambitious, the authors would not need to state anything about representativeness of the shown image, but can just demonstrate it with data.

A similar observation can be made about the CLEM data in Fig. 6a.

We thank the reviewer for this suggestion. We have added additional CLEM examples in Fig. 1a and in the new Fig. 6d (formerly Fig. 6a) and have included the number of biological replicates ($n = 3$). We have retained the term "representative" in the figure legends while explicitly indicating the number of independent biological repeats.

In the section "AL are highly dynamic and can fuse with the NE...", its title, main text and figure labelling: Fig. 2e and SI videos 2,3 are said to show that there are fusion and fission events of AL foci with the NE. The shown images, even for Z slices in confocal microscopy, at this scale still represent 2D projections of a 3D reality, which may complicate interpretation. Is it obvious, depending on the depth resolution in these experiments, that the foci are not drifting behind/in front of the view of the nucleus?

We thank the reviewer for raising this important point. Although we occasionally observed foci drifting behind or in front of the nuclear plane at low frequency, we minimized this

possibility by acquiring smaller Z-stacks (3–5 μm) rather than the standard 10 μm . Importantly, none of the examples or videos presented in the manuscript show foci drifting behind or in front of the nucleus. In addition, during quantification, potential overlapping foci were excluded from the analysis rather than included as insertion or fusion events.

What data is the quantification in Fig. 2d based on? Having looked at Fig. 2e and SI videos 2,3, I struggle to identify the numbers of attachment/detachment events noted in Fig. 2d, and from the underpinning data shown, it seems that their identification is not that unambiguous. Could the authors explain more explicitly how this (per se very welcome) quantification results from the microscopy data?

We thank the reviewer for this comment. The quantification shown in Fig. 2d is based on fast live-imaging microscopy data, with Supplementary Video 3 providing a representative example. Quantification was performed by focusing specifically on regions adjacent to the nuclear envelope (NE). Because recording attachment and detachment events across the entire nuclear circumference was not feasible due to the limited field of view, each nucleus was divided into multiple NE-adjacent sections. Foci dynamics were then tracked frame by frame within each section, and attachment and detachment events were recorded accordingly. These events were subsequently pooled for statistical analysis. A detailed description of the image acquisition and quantification procedure has been added to the Methods section.

Data in Fig. 2f are shown to further substantiate the claim of AL-insertion into NE, albeit with the authors' caveat about long-term stability and brightness of the fluorophores in these data. Is there a negative control for these experiments?

We thank the reviewer for this comment. We have included a -40 s time point prior to photoconversion, which does not show an increase in nuclear signal at -20 s. In contrast, 20 s after photoconversion, a statistically significant increase in the red fluorescent signal is observed. This pre-photoconversion time point serves as an internal negative control for the experiment.

Minor:

Fig. 1b, top-left panel: A "2" label appears to be missing in this image.

We thank the reviewer for pointing this out. The missing label has now been added.